

# Quantification of water vapour transport from the Asian monsoon to the stratosphere

Matthias Nützel[1], Aurelien Podglajen[2], Hella Garny[1,3], and Felix Ploeger[2,4]

[1]Deutsches Zentrum für Luft- und Raumfahrt, Institut für Physik der Atmosphäre, Oberpfaffenhofen, Germany
[2]Institute of Energy and Climate Research: Stratosphere (IEK-7), Forschungszentrum Jülich, Jülich, Germany
[3]Meteorological Institute, Ludwig Maximilians Universität, Munich, Germany
[4]Institute for Atmospheric and Environmental Research, University of Wuppertal, Wuppertal, Germany

**Correspondence:** Matthias Nützel (matthias.nuetzel@dlr.de)

**Abstract.**

Numerous studies have presented evidence that the Asian summer monsoon anticyclone substantially influences the distribution of trace gases – including water vapour – in the upper troposphere and lower stratosphere (e.g. Santee et al., 2017). Stratospheric water vapour in turn, is strongly affecting surface climate (cf. e.g. Solomon et al., 2010). Here, we analyse the

characteristics of water vapour transport from the upper troposphere in the Asian monsoon region to the stratosphere employing a multiannual simulation with the chemistry-transport model CLaMS (Chemical Lagrangian Model of the Stratosphere). This simulation is driven by meteorological data from ERA-Interim and features a water vapour tagging that allows us to assess the contributions of different upper tropospheric source regions to the stratospheric water vapour budget. Our results complement the analysis of air mass transport through the Asian monsoon anticyclone by Ploeger et al. (2017). The results show that the

transport characteristics for water vapour are mainly determined by the bulk mass transport from the Asian monsoon region. Further, we find that, although the relative contribution from the Asian monsoon region to water vapour in the deep tropics is rather small (average peak contribution of 14% at 450 K), the Asian monsoon region is very efficient in transporting water vapour to this region (when judged according to its comparatively small spatial extent). With respect to the Northern Hemisphere extratropics, the Asian monsoon region is much more impactful and efficient regarding water vapour transport than e.g.

the North American monsoon region (averaged maximum contributions at 400 K of 29% vs. 6.4%).

## 1 Introduction

Atmospheric water vapour is a key greenhouse gas (e.g. Held and Soden, 2000; Schmidt et al., 2010; Müller et al., 2016). Despite the extremely low average water vapour mixing ratios in the stratosphere (considerably below $10\,\mu\mathrm{mol\,mol^{-1}}$ cf. e.g.

Hegglin et al., 2013, their Fig. 4) compared to tropospheric abundances (cf. e.g. Sherwood et al., 2010, their Fig. 2), changes in stratospheric and tropical tropopause layer (TTL; Fueglistaler et al., 2009) humidity can noticeably impact Earth's surface



climate (e.g. Solomon et al., 2010; Riese et al., 2012). Additionally, changes in water vapour can alter stratospheric chemistry and hence the abundances of other radiatively active trace gases (e.g. Stenke and Grewe, 2005), which leads to a secondary radiative effect associated with water vapour changes (e.g. Dvortsov and Solomon, 2001). Based on findings e.g. by Brewer (1949) and the current understanding of the Brewer–Dobson circulation (BDC; Butchart, 2014) most of the air masses and

thus also water vapour are expected to enter the stratosphere via the tropics. As proposed by Brewer (1949), TTL temperatures hence strongly influence water vapour abundances in the stratosphere. This can be seen by the so-called water vapour tape recorder (Mote et al., 1996, see also Fig. 3), i.e. a seminannual seesaw of positive and negative water vapour anomalies that ascends in the tropical pipe (Plumb, 1996) and is related to the seasonal cycle of TTL temperatures (e.g. Yulaeva et al., 1994).

In addition to the tropical pathway to the stratosphere, several studies have argued that the Asian monsoon region (including the Tibetan Plateau) and the associated anticyclone might be a preferred transit region for air masses – which are imprinted with enhanced signatures of trace gases with mainly tropospheric origin (see e.g. Santee et al., 2017; Lelieveld et al., 2018) – from the troposphere to the stratosphere (e.g. Dethof et al., 1999; Fu et al., 2006; Lelieveld et al., 2007; Randel et al., 2010; Ploeger et al., 2017). As an example, Dethof et al. (1999) have argued that the Asian monsoon may influence the water vapour budget

of the extratropical lower stratosphere (LS). Further, Randel et al. (2010) have provided evidence that air masses from the upper troposphere-lower stratosphere (UTLS) in the Asian monsoon region can be transported to the tropical LS from where these air masses can ascend further into the stratosphere. A recent study by Yu et al. (2017) showed an important influence of the Asian summer monsoon anticyclone on the aerosol budget in the stratosphere of the Northern Hemisphere (NH) and pointed out the efficiency of this circulation system in contributing to the aerosol budget.

The impact of water vapour which is associated with the Asian monsoon (anticyclone) on the stratospheric water vapour budget has been investigated in several studies e.g. Bannister et al. (2004), Gettelman et al. (2004), Fu et al. (2006), Lelieveld et al. (2007), Kremser et al. (2009) and Wright et al. (2011) – partly leading to apparently different results. The study by Bannister et al. (2004), which is based on data from an atmospheric general-circulation model and additional offline calculations,

investigated the tropical water vapour tape recorder signal and in particular the influence of the Asian monsoon on tropical stratospheric water vapour. In this study, Bannister et al. (2004) found that a region over the Central Pacific, which is associated with air masses from the Asian monsoon, clearly contributes to the wet phase of the tape recorder signal. Using different reanalysis data sets and satellite cloud data, Wright et al. (2011) assessed the moistening effect of Asian monsoon convection in the tropical stratosphere at 68 hPa during NH summer. Seemingly contradictory to Bannister et al. (2004), Wright et al. (2011)

concluded from their analysis that the Asian monsoon has only a weak moistening effect.

With respect to transport of idealised tracers to the stratosphere, studies e.g. by Orbe et al. (2015), Garny and Randel (2016), Pan et al. (2016) and Ploeger et al. (2017) investigated the transport pathways from the Asian monsoon region to the stratosphere and quantified the impact of the Asian monsoon region on the stratospheric air mass budget. In particular,

using the chemistry-transport model CLaMS (Chemical Lagrangian Model of the Stratosphere; McKenna et al., 2002), Ploeger



et al. (2017) recently found that air masses from the core and the edge region (cf. their sensitivity analysis) of the anticyclone typically cross the tropopause vertically and are transported horizontally to the tropics and NH extratropics afterwards. Further they showed, that these air masses are subsequently transported in accordance with the broad scale structure of the BDC, i.e. upwards in the tropics and downwards in the extratropics, and that the idealised Asian monsoon air mass tracer mimics the transport characteristics of a tropospheric trace gas, which is affected by the Asian monsoon. The transport characteristics of this trace gas have been previously described in Randel et al. (2010).

With respect to water vapour transport, freeze-drying (e.g. of air masses encountering low temperatures in the southern part of the monsoon anticyclone; see e.g. Fueglistaler et al., 2005; Wright et al., 2011) might noticeably influence the transport characteristics from the Asian monsoon region to the stratosphere. Hence, the conclusions of Ploeger et al. (2017) obtained for inert tracers might not be directly transferable to water vapour transport. Consequently, this study investigates if the transport characteristics from the Asian monsoon anticyclone to the stratosphere change, when a tracer like water vapour, which critically depends on the temperature field, is considered. The intricate relationship between water vapour, temperature and convection was highlighted e.g. by Randel et al. (2015). They showed that for LS water vapour in the Asian monsoon region, the temperature change of the UTLS due to convection (more vigorous deep convection corresponding to lower temperatures on the south-eastern side of the Asian monsoon anticyclone and thus a drier LS) is a key process, which outweighs the possible moistening effect associated with convection, which has for example been observed in the upper troposphere (UT) Asian monsoon region (Randel and Park, 2006, their Fig. 8). This finding is in agreement with trajectory calculations and satellite observations of convection in the Asian monsoon region presented by Wright et al. (2011), which showed that the bulk of convectively influenced air masses from the Asian monsoon region is detrained below the tropopause (c.f. their Fig. 5 and corresponding discussion).

Considering the importance of stratospheric water vapour and the aforementioned potential of the Asian monsoon for influencing the stratospheric water vapour budget, we will quantify the contribution of the Asian monsoon on stratospheric water vapour in this study. In detail the main goals of this study are: 1) To highlight the transport pathways of water vapour from the UT in the Asian monsoon to the stratosphere and to contrast air mass and water vapour transport from the Asian monsoon region to the stratosphere. 2) To quantify the impact of the Asian monsoon on the stratospheric water vapour budget. 3) To compare the water vapour and mass transport (efficiency) from the Asian monsoon to the transport (efficiency) from additional source regions, such as the North American monsoon and the entire tropics. For this we will employ multiannual model results from CLaMS driven by observationally constrained reanalysis data. The employed simulation allows to tag mass and water vapour according to different source regions. These tagged water vapour and mass tracers are used to derive a source region attribution for water vapour and air masses throughout the stratosphere. Although water tagging has often been used to study the water cycle and disentangle the contributions of different water sources in the troposphere (e.g. Koster et al., 1992; Bosilovich and Schubert, 2002), this method has – to the best of our knowledge – never been applied to the stratospheric water vapour budget. The water tagging is particularly suited for our purpose here, since it allows a decomposition of water origins consistent with the model treatment of water transport and removal through freeze-drying.



The manuscript is structured as follows: In Sect. 2 the employed model and data along with the water vapour tagging method are described. Section 3 contains a short evaluation of our model results along with the assessment of transport of water vapour from the Asian monsoon region to the stratosphere using tagged water vapour. Finally, Sect. 4 and Sect. 5 contain the discussion of our results and our conclusions, respectively.

## 2   Data and method

### 2.1   Model data

The transport of water vapour and mass tracers is assessed using a simulation performed with CLaMS (McKenna et al., 2002) in its 3D version (Konopka et al., 2004). This simulation covers the period January 2010 to December 2014. CLaMS treats advective transport in a Lagrangian way, following the (diabatic) trajectories of the air parcels computed using the wind fields and heating rates from the European Centre for Medium-Range Weather Forecasts (ECMWF) reanalysis ERA-Interim (Dee et al., 2011). For the simulations presented here, ERA-Interim data at $1° \times 1°$ resolution was employed. A special feature of CLaMS is its mixing parameterization, which represents small-scale mixing, which is not resolved in the reanalysis data. Mixing in CLaMS is driven by the resolved deformation of the flow (Konopka et al., 2004; Riese et al., 2012, cf. also Fig. 2 of the latter). With this parameterization, different air parcels can be merged and new air parcels can be inserted; this also contributes to transport, especially on the vertical (Konopka et al., 2007), in addition to the mean transport, which is resolved by the reanalysis flow.

Regarding the water vapour field, the model includes a simple freeze-drying parameterization. Two water variables, water vapour and ice water content, are treated by the module. At each model time step, the vapour in excess of saturation (saturation water vapour pressure calculated using the formula of Marti and Mauersberger, 1993, with the temperature taken from ERA-Interim) is condensed into ice. On the contrary, if the air is sub-saturated and ice exists, it is sublimated to maintain saturation. The ice phase is then depleted by sedimentation, represented as a fall-out of ice: assuming a mean radius for the ice crystals, a fallspeed and the corresponding fallen path during the duration of the time step are calculated. The fallen path is then compared to a characteristic sedimentation length (300 m, optimised by Ploeger et al., 2013), and the fraction of ice that has fallen more than that characteristic length is removed from the air parcel. For an improved water vapour budget in the stratosphere methane oxidation is also accounted for (cf. Ploeger et al., 2013). Below about 500 hPa, the water content is set to that of ERA-Interim. For the described analyses gridded data based on monthly means is considered.

### 2.2   Tagging-method

In the simulation we included two types of tracers: 1) Tracers that are inert and are initialised with unity in specific source regions during specific time periods and are only transported (also referred to as mass tracers in the following). 2) Total water tracers which are initialised with the total water of specific source regions during specific time periods and undergo sedimentation loss besides transport. The sedimentation loss in turn is influenced by freeze-drying. Both tracer sets are initialised with



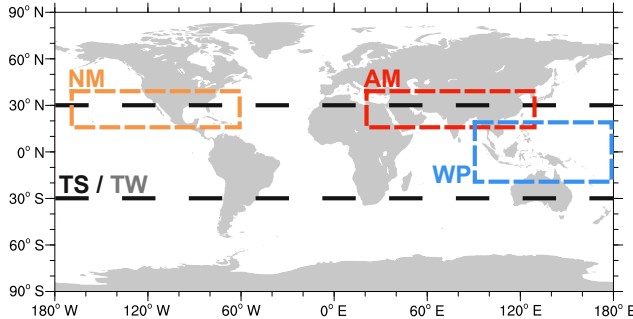

**Figure 1.** Definition of the source regions used for the initialisation of mass and water vapour tracers. According to the tagging region and period, the tracers are named as follows: **AM** (**A**sian summer **m**onsoon, 15°N–40°N x 20°E–130°E, red dashed), **NM** (**N**orth American **m**onsoon, 15°N–40°N x 170°W–60°W, orange dashed), **TS/TW** (**t**ropics NH **s**ummer/**w**inter, 30°S–30°N, black dashed) and **WP** (**w**arm **p**ool, 20°S–20°N x 90°E–180°E, blue dashed). For the size, tagging period and tagging height of the tracer regions see Table 1.

zero at the beginning of the simulation period and evolve "freely" (details on the evolution of the tracers are presented later) outside of the corresponding source regions. According to the model workflow, the tagging within the respective source regions is performed every 24 h in analogy to the tagging methods in Vogel et al. (2016) and Ploeger et al. (2017). The source regions of this study are defined and illustrated in Fig. 1. In the vertical, the source regions lie within the layer of 370–380 K and the

tagging initialisation is performed during July–August (JA) for summer tracers and during January–February (JF) for winter tracers. The winter tracers are included to provide complementing estimates of transport from the tropics and the warm pool region to the stratosphere during NH winter. The details of the source regions and tagging periods are summarised in Table 1. During transport these tagged air masses are mixed. Finally, at a specific time and location the concentration of the mass tracer gives the relative contribution of that source region and tagging period to the air mass at that point (cf. e.g. Orbe et al., 2013,

2015). Similarly, the water tracer shows the total water associated with a specific source region and tagging period. Note, that the model's methane oxidation does not influence the total water tracers outside of the tagging regions, i.e. there is no positive tendency for tagged water outside of the tagging regions due to methane oxidation. For the mass tracers the signal from the previous year's initialisation is removed directly before the start of a new inert tracer initialisation, i.e. the signal shows only contributions of the air mass tracer up to at most one year after release of the tracer. In contrast, the total water tracer is not

reinitialised, i.e. it is possible for water vapour tracers from different years to accumulate.

In a similar way Ploeger et al. (2017) have used air mass tagging to investigate air mass transport from the Asian monsoon anticyclone to the stratosphere. In contrast to their investigation we focus on the transport of water vapour. Further, we do not limit the impact of the Asian monsoon to the core of the anticyclone. In Ploeger et al. (2017) only air masses within the anticyclone, which was defined using a potential vorticity based border (cf. Ploeger et al., 2015), have been tagged. Here, we

simply use a box region (cf. Fig. 1) to tag air masses and attribute them to a specific source region. We do this as using only the core of the monsoon anticyclone might be too stringent as air masses that are transported (and hence affected) by the strong anticyclonic winds on the border of the anticyclone can also be transported to the extratropical lower stratosphere (e.g. Vogel



**Table 1.** Details regarding tagged regions, height range and time periods of tracer initialization for the different source regions. The area fraction is given in per cent relative to the tropics (30°S–30°N).

| Source region | Latitude | Longitude | Height range | Period | Area fraction |
|---|---|---|---|---|---|
| AM | 15°N–40°N | 20°E–130°E | 370–380 K | JA | 12.09% |
| NM | 15°N–40°N | 170°W–60°W | 370–380 K | JA | 12.09% |
| TS | 30°S–30°N | 180°W–180°E | 370–380 K | JA | 100% |
| TW | 30°S–30°N | 180°W–180°E | 370–380 K | JF | 100% |
| WP | 20°S–20°N | 90°E–180°E | 370–380 K | JF | 17.39% |

et al., 2014). Further, Garny and Randel (2016) have conducted sensitivity studies with a trajectory model which indicate that the qualitative results for transport from the UT in the Asian monsoon region to the stratosphere do not critically depend on the initialisation of the trajectories in the Asian monsoon region (cf. e.g. their Fig. 17).

For technical reasons instead of water vapour, total water is being tagged and followed along the parcels pathway. The mixing of the tagged total water tracers is performed in analogy to the mixing of normal compounds, e.g. water vapour. The evolution of tagged total water of a source region $i$ (with $i \in \{AM, NM, TS, TW, WP\}$) due to the fallout of the ice from time step $(t)$ to $(t+1)$ on a certain trajectory is represented as follows:

$$TWC^i(t+1) = TWC^i(t) \times \frac{TWC(t+1)}{TWC(t)}, \tag{1}$$

where $TWC = IWC + H_2O$ denotes the total water content (TWC), i.e. water vapour ($H_2O$) plus ice water content (IWC) and the superscript $i$ refers to the respective quantities from this specific source region.

From the total water tracer the absolute contribution of that source region $(i)$ to the water vapour budget at a certain location $(H_2O^i)$ can be approximated via:

$$H_2O^i = H_2O \times \frac{TWC^i}{TWC} = H_2O \times \frac{TWC^i}{IWC + H_2O}. \tag{2}$$

This simplification is expected to yield reasonable results as the amounts of IWC in the stratosphere are quite small, i.e. $TWC = IWC + H_2O \approx H_2O$. It is further noted that – as is already obvious due to overlapping source regions (cf. Fig. 1) – the water vapour and mass contributions of different source regions are not exclusive, i.e. a parcel may be tagged several times (without loosing previous tags) in different source regions (e.g. when the parcel passes through the intersecting regions of TS and AM, but also if a parcel crosses different source regions successively).

## 2.3 Observations

Since our diagnostic of regional water contributions to the stratosphere relies on the water vapour representation of the model, it is necessary to compare the simulated water vapour with observations. Here for the evaluation of CLaMS water vapour in



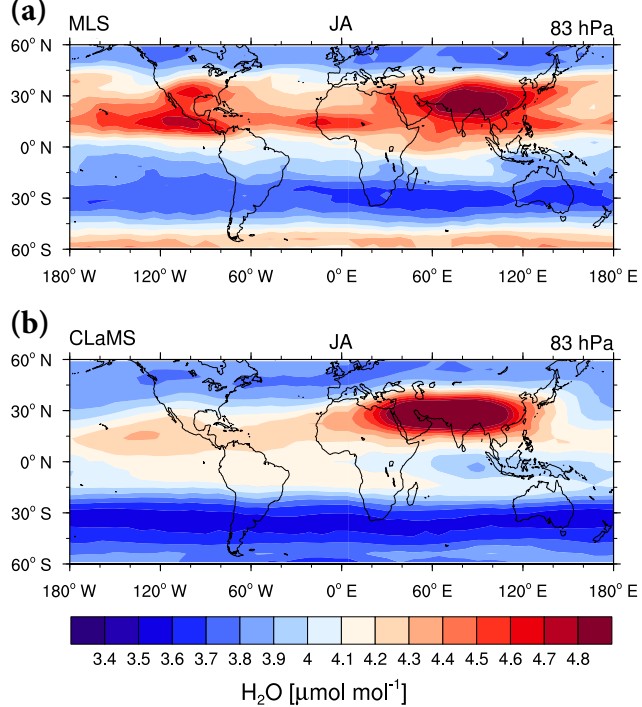

**Figure 2.** Maps of $H_2O$ ($\mu$mol mol$^{-1}$) during JA 2010–2014 at 83 hPa from **(a)** MLS and **(b)** CLaMS data, which was processed to enable the comparison with MLS data (see text and cf. Sect. 2.3).

the UTLS we use data from MLS (Microwave Limb Sounder; Waters et al., 2006). MLS provides measurements of a variety of atmospheric trace gases (including water vapour), which have been commonly used in studies, which focus on the UTLS in the Asian monsoon region (cf. Santee et al., 2017). Here, MLS water vapour version 4.2 data is employed (Lambert et al., 2015) and details on data quality, suggested data pre-processing etc. of this data version are presented in Livesey et al. (2018). The

5 MLS water vapour product of a previous data version has been described and evaluated by Read et al. (2007) and Lambert et al. (2007). In the following ClaMS data will be evaluated against water vapour data from MLS. In the corresponding analyses, we will show CLaMS data, which was sampled at the MLS measurement locations and smoothed via the MLS averaging kernels (cf. Livesey et al., 2018, for an overview, instruction details and a reference regarding MLS averaging kernel data). The employed procedure for processing CLaMS data to facilitate the comparison with MLS data is described in Ploeger et al.

10 (2013).



## 3  Results

### 3.1  Modelled and observed water vapour

Previous comparisons of ERA-Interim based CLaMS results with satellite data have demonstrated the model's capability to simulate transport in the UTLS in the Asian monsoon region (e.g. Vogel et al., 2016; Ploeger et al., 2017). Comparisons of MLS and CLaMS water vapour fields during June–August (JJA) and December–February (DJF) at 380 K have been previously presented, e.g. in Poshyvailo et al. (2018; see e.g. their Fig. 5). These comparisons show a reasonable agreement of simulated and observed water vapour fields and especially, the water vapour signals of the Asian and North American monsoon regions during NH summer are present in the modelled water vapour distributions (Poshyvailo et al., 2018). Nevertheless, previous studies employing CLaMS, e.g. by Ploeger et al. (2013) and Poshyvailo et al. (2018), also showed that there are some discrepancies e.g. regarding the strength of the water vapour signal from the Asian in comparison to the North American monsoon and the absolute values of water vapour in the Asian monsoon region. However, as noted by Poshyvailo et al. (2018) direct reanalysis $H_2O$ values from ERA-Interim and JRA-55 (Japanese 55-year Reanalysis; Kobayashi et al., 2015) show stronger deviations from MLS satellite observations than CLaMS simulated water vapour (cf. their Fig. 17).

Figure 2 shows maps of water vapour at 83 hPa during JA, i.e. during the initialisation period of the summer tracers. The two panels of this figure display MLS data and CLaMS data that was sampled at MLS measurement locations and convolved with the MLS averaging kernels. In agreement with MLS data, CLaMS data show enhancements of water vapour in the Asian and North American monsoon region at 83 hPa with higher absolute values in the Asian monsoon region than in the North American monsoon region. However, the relative strength of the Asian summer monsoon signal compared to the water vapour signal over the North American monsoon region is more pronounced in CLaMS (bottom panel) than in MLS data (in agreement with a relatively stronger Asian monsoon water vapour signal in CLaMS than in MLS data on the 380 K potential temperature surface in Poshyvailo et al., 2018, cf. their Fig. 5). Below the tropopause the Asian monsoon shows a clearer water vapour maximum than the North American monsoon region also in MLS data (cf. e.g. Fig. 1 of Heath and Fuelberg, 2014). Further, MLS data show a weak maximum over the African monsoon region and an additional secondary maximum over the North American continent, which is not visible in CLaMS data. Additionally, the minimum water vapour in the SH tends to be located farther south in CLaMS than in MLS data, which also show a stronger gradient in the south. Partly, these differences might arise from uncertainties in the satellite product Livesey et al. (2018). Further, we note that we did not include the apriori profiles, which might increase the agreement between satellite and model data. Another explanation for the differences over the North American continent might be the lack of explicit convection in CLaMS. In observational data this region is often influenced by mesoscale convective events which might lead to a moistening (cf. eg. Huntrieser et al., 2016, and references therein) that might be lacking in CLaMS. Similarly, summertime convection over the African continent (cf. e.g. Huntrieser et al., 2011, and references therein), which is not fully captured in CLaMS might be responsible for the weak water vapour signal over the African monsoon region in comparison with satellite data.



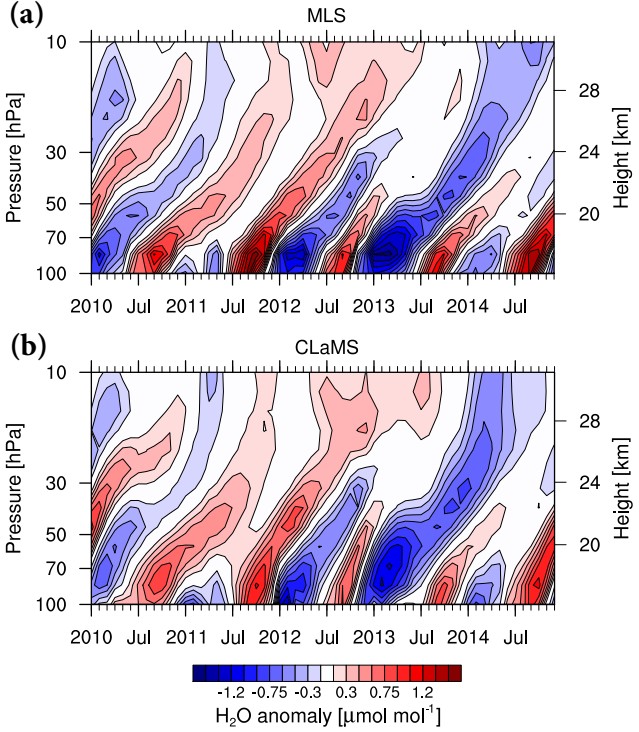

**Figure 3. (a)**: $H_2O$ tape recorder signal from MLS data (in $\mu mol\, mol^{-1}$). The signal has been calculated as the zonally averaged temporal anomaly of $H_2O$ for the years 2010–2014 averaged over $10°$S-$10°$N. Dry phases are shown in blue and wet phases in red. **(b)**: As in **(a)** but for CLaMS data, which was processed for the comparison with MLS data (cf. Sect. 2.3).

The discrepancies between model and satellite data, e.g. the representation of the relative strength of the water vapour signal in the Asian monsoon compared to the North American monsoon region, might influence our results with respect to the contribution of water vapour from the Asian monsoon (e.g. in comparison with the North American monsoon) region to stratospheric water vapour. Here, we point out the apparent complexity of modelling water vapour in the TTL region as can be for

5    example seen from differences in water vapour fields between modern reanalyses (cf. Fig. 17 in Poshyvailo et al., 2018) due to various interacting processes (e.g. representation of convection, largescale transport, freeze-drying). Hence, modelling realistic water vapour distributions, in particular in the Asian monsoon region, is challenging (cf. e.g. Wang et al., 2018, and references therein). Nevertheless, the sensitivity of our results to this issue will be further assessed in Sect. 4.

10    As an additional evaluation of the representation of water vapour transport in CLaMS driven by ERA-Interim data, the water vapour tape recorder signal (cf. Mote et al., 1996) is shown in Fig. 3a and Fig. 3b for MLS and CLaMS data. These figures display the anomalies with respect to the 2010–2014 mean water vapour profile over the region $10°$S-$10°$N. Up to minor differences the tape recorder signals from MLS and CLaMS data show excellent agreement both in the timing and strength of the $H_2O$ anomalies in the stratosphere. In particular, the occurrence of the driest phase in 2013 and the comparatively weak



signal of the dry phases for 2011 and 2014 (all at ∼80–50 hPa) are captured in the model data. These two NH winters are influenced by the westerly phase of the quasi-biennial oscillation (QBO; see e.g. Newman et al., 2016; Osprey et al., 2016, for a QBO time series covering the simulation period described here). The QBO is known to influence stratospheric and tropopause temperatures (see e.g. Randel et al., 2000; Baldwin et al., 2001, and references therein) and hence, it also modulates strato-

spheric water vapour (e.g. Giorgetta and Bengtsson, 1999, who assess the QBO impact on lower stratospheric water vapour in a model study). Thus, the QBO phase can explain the comparatively weak water vapour anomalies during 2011 and 2014 (cf. e.g. Fig. 3b of Diallo et al., 2018). The anomalously high cold point tropopause temperatures for the NH winter season 2010–2011 can be found in Kumar et al. (2014; cf. their Fig. in the Annex A).

In summary, the presented results show that CLaMS is suitable for modelling water vapour in the LS and for assessing water vapour transport within the tropical pipe. In particular, the model's capability of reproducing satellite-based water vapour variability on seasonal and interannual time-scales in the tropical stratosphere has been shown. Nevertheless, there are also deficiencies of the model in reproducing absolute values of water vapour. After the presentation of our results regarding water vapour transport from the Asian monsoon to the stratosphere in the following, the sensitivity of these results to this issue will

be addressed in the discussion (Sect. 4).

### 3.2 Transport pathways of Asian monsoon water vapour to the stratosphere

The contribution of the mass tracer from the Asian monsoon region (AM mass tracer; as fraction of the total air mass, colour-coded) and overlaid the contribution of the corresponding $H_2O$ tracer (contours) to the respective zonal temporal mean water

vapour over the course of a year is shown in Fig. 4. The corresponding analysis for the tropics during NH summer (TS mass and water vapour tracers) is shown in Fig. 5. The presented climatologies for the summer tracers are based on data from July–September (JAS) 2010 to April–June (AMJ) 2014 (i.e. simulation months 7–54 for NH-summer tracers), as during the first six months of the simulation the summer tracers have not been initialised yet. Hence, these months were excluded and consequently the climatologies represent JAS 2010–2013, October–December (OND) 2010–2013, January–March (JFM) 2011-2014 and

AMJ 2011–2014.

First, we focus on air mass transport from the Asian monsoon region and the tropics during NH summer (colour coding in Figs. 4 and 5, respectively). For the AM mass tracer, the main characteristics are as follows: During JAS air masses from the Asian monsoon region are transported rapidly across the mean local tropopause in the latitude range of the initialisation region, which should be mostly located below the tropopause (cf. Ploeger et al., 2017, their Fig. 7). Subsequently, during OND,

the AM air mass tracer splits and one fraction of the AM air mass tracer is transported to the tropics (from OND to JFM). This air mass later experiences upward transport in the tropical pipe during JFM and AMJ. The second large fraction of the AM air mass tracer is located in the extratropical LS (mostly below ∼400 K) during OND and experiences downward motion during JFM and AMJ. This qualitative behaviour is in agreement with the transport of air masses by the BDC and with the description of transport from the Asian monsoon anticyclone tracer to the stratosphere presented in Ploeger et al. (2017, cf.



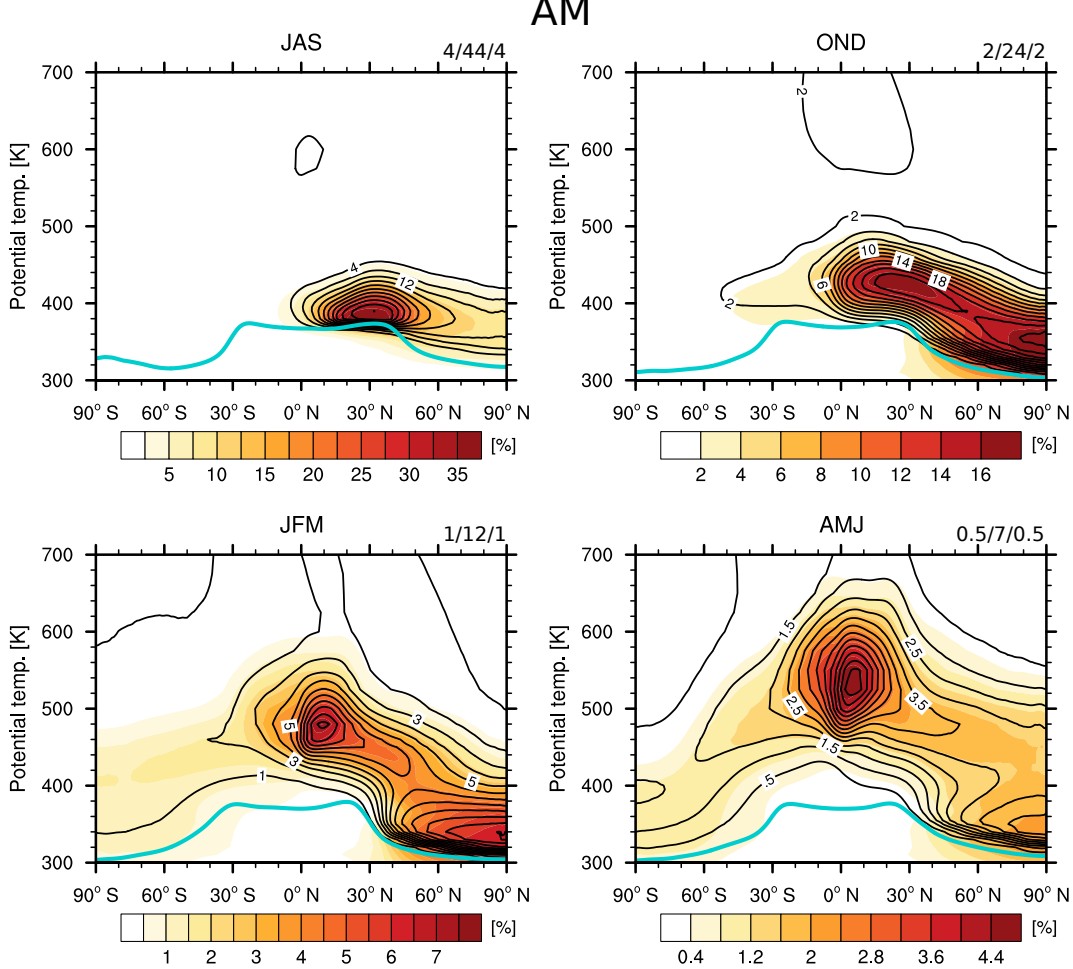

**Figure 4.** Climatology of mean mass (colour-coded) and relative water vapour (contours) contribution of the AM tracer (both in %) over the course of a year. Note, that each season features an individual colourbar. Numbers in the top right corner of each panel indicate the contour line spacing as min/max/delta. The light blue lines represent the mean WMO-tropopause based on ERA-Interim data. The main characteristics of mass transport from the AM tracer are similar to transport from the core of the Asian monsoon anticyclone as displayed in Fig. 1 of Ploeger et al. (2017).

their Fig. 1). The main differences between the anticyclone air mass tracer transport presented in Ploeger et al. (2017) and the transport of the AM mass tracer presented here are restricted to differences in the absolute contribution of the Asian monsoon air masses, which are related to the difference in the initialisation regions between the two studies (cf. Sect. 2.2). This shows that the pathways of air masses from the UT (here, the 370–380 K initialisation layer) in the Asian monsoon region are not

5    sensitive to the exact initialisation of the tagged air masses, i.e. the initialisation within a box region or within the monsoon anticyclone defined by a PV-boundary – as in Ploeger et al. (2017) – yield similar qualitative results. This is in accordance with the sensitivity study presented in Ploeger et al. (2017), which analysed transport from the edge of the anticyclone and with



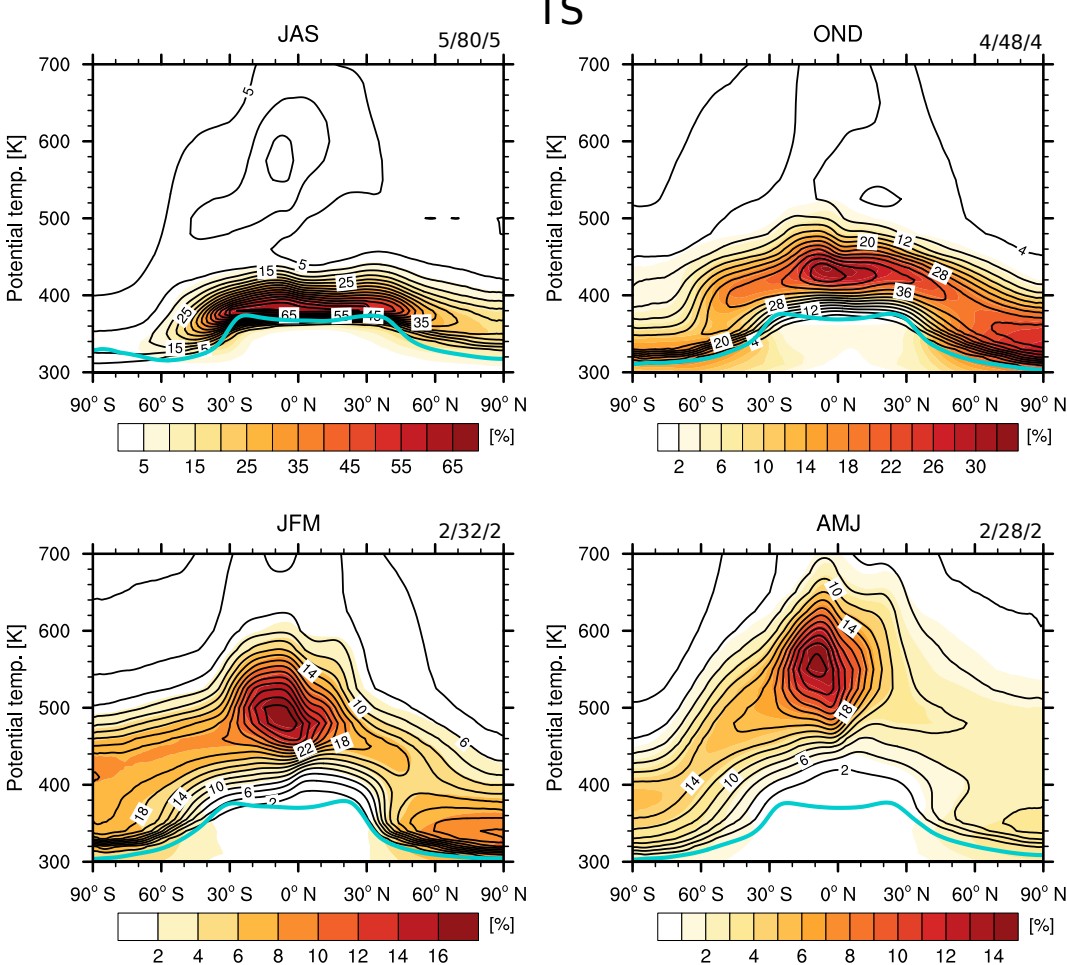

**Figure 5.** As in Fig. 4 but for the TS tracer.

the sensitivity study described in Garny and Randel (2016), which was focused on the impact of the horizontal distribution of starting position of trajectories in the UT in the Asian monsoon region on the qualitative transport characteristics.

The TS mass tracer shows a similar qualitative behaviour as the AM mass tracer. However, as the TS mass tracer is initialised also in the Southern hemisphere (SH) a considerable fraction of the tropical air masses has already been transported to the SH extratropics in OND. The absolute values of the mass contribution from the tropics tracer are considerably higher than the contributions from the Asian monsoon tracer in accordance with the larger area of the tropics.

Now, we turn to the investigation of water vapour transport from the AM (and TS) region to the stratosphere, which has not been previously investigated in Ploeger et al. (2017). For the relative water vapour contribution both the AM and the TS water vapour tracers show maximum contributions to the water vapour budget that are co-located with the maximum mass contributions, i.e. the main pathways and contributions of tagged water vapour are mainly determined by the main mass trans-





port pathways. It is noted that there are non-zero water vapour contributions during JAS from the previous tagging periods of the tracers (e.g. the local maximum at roughly 540–600 K at the Equator) as the water vapour tracers are not reinitialised in contrast to the mass tracers (cf. Sect. 2.2). Also, for both regions, the mean relative contribution to water vapour is typically higher than the mean mass contribution. As can be seen from the relation during the first year of the simulation (see time series

plots in Sect. 3.3), this cannot be explained by the reinitialisation of the mass tracers in comparison to the water vapour tracers (cf. Sect. 2.2).

### 3.3  Evolution of Asian monsoon mass and water vapour contributions in the tropics and the NH extratropics

Figure 6a shows the temporal evolution of the absolute water vapour mixing ratios at 450 K averaged over 10°S–10°N for the

individual source regions and for the total water vapour (dark blue line). The corresponding relative contributions of the mass (dashed lines) and water vapour (solid lines) tracers at 450 K averaged over 10°S–10°N are shown in Fig. 6b. In these figures, besides the tropics tracer released during NH summer (TS, black lines) and the Asian monsoon tracer (AM, red lines), also the tropics tracer released during NH winter (TW, grey lines), the North American monsoon tracer (NM, orange lines) and the Warm Pool (WP, light blue lines) tracer results are shown. With respect to the total water vapour contribution, the tropical

tracers TW and TS show the highest contributions with peak values as high as $\sim 1.6~\mu\text{mol mol}^{-1}$ and $\sim 2.6~\mu\text{mol mol}^{-1}$, respectively.

As can be expected, also the air mass contributions to the tropics at 450 K from the tropical source regions (TS and TW) are the largest. Further, the TW mass tracer shows higher peak contributions (up to 78%, and higher variability) than the TS mass tracer (at most ∼43%). This is in qualitative agreement with the seasonal cycle of the strength of the tropical upwelling

in ERA-Interim (cf. e.g. Abalos et al., 2012, their Fig. 3). The time lag of the maximum of the signal arrives approximately 3–5 months after the start of the initialisation. This is in accordance with the slow upward movement of air masses in the BDC within the tropical pipe (cf. e.g. the slow upward transport of trajectories within the tropical stratosphere by the residual circulation as displayed in Fig. 2 of Birner and Bönisch, 2011). This time lag seems to be slightly reduced for the winter tracers compared to the summer tracers, in agreement with the seasonal cycle of the BDC.

During the simulated years the mass contribution of the Asian monsoon tracer reaches at maximum 15%. The average peak relative contributions to stratospheric water vapour at 450 K in the deep tropics for the individual water vapour tracers TS, TW, AM, NM and WP are roughly 51, 63, 14, 6.7 and 24%, respectively (cf. Table 2). For the TS and the AM tracers the water vapour contributions are typically higher than the respective mass contributions. This can be expected as air from these regions

is supposed to be relatively moist, i.e. featuring higher water vapour mixing ratios than the average air masses and hence, the water vapour contribution should be higher than the mass contribution. In contrast, for the WP tracers the relation is reversed, which fits to relatively dry air originating from this region. Although, one might expect that during NH winter the contribution of water vapour compared to mass from the tropics (TW tracers) should be lower, as well, the relation is not as clear as for the WP tracers. In particular, in 2011 and 2014 the water vapour contribution is higher than the mass contribution. This arises most



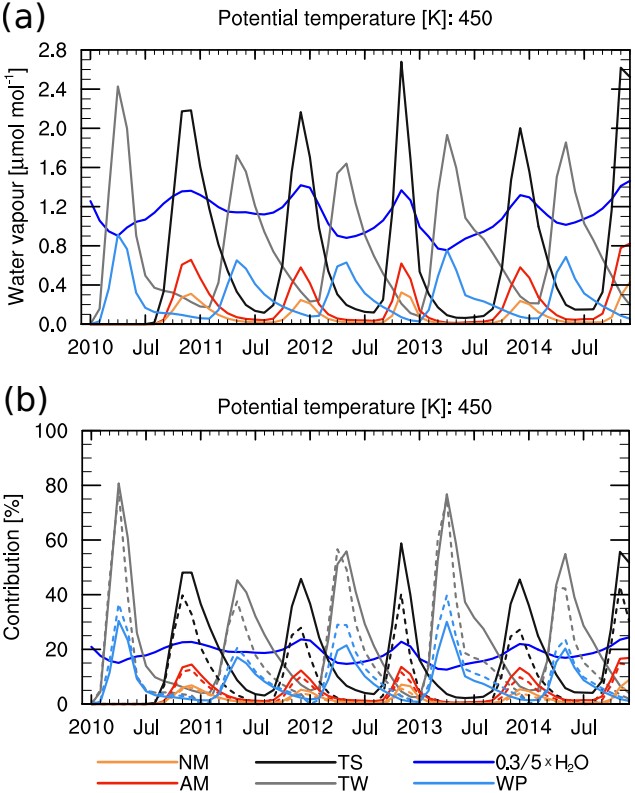

**Figure 6. (a)**: Time series of the $H_2O$ mean mixing ratios from the different source regions and water vapour (all in $\mu mol\, mol^{-1}$, total $H_2O$ scaled by $\times 0.3$). The time series were calculated over the region $10°S$–$10°N$ at 450K for the source regions TS (black), TW (grey), WP (light blue), AM (red) and NM (orange). Total water is shown in dark blue. **(b)**: Time series of the percentage contribution of the mass (dashed) and $H_2O$ (solid) tracer for the different source regions averaged over $10°S$–$10°N$ at 450 K (total $H_2O$ scaled by $\times 5$, where 1% equals $1 \mu mol\, mol^{-1}$). Colour coding as in **(a)**.

likely because those two years show relatively high tropical cold point tropopause temperatures (not shown), which are probably related to the westerly phase of the QBO during these years (cf. Sect. 3.1; Baldwin et al., 2001, and references therein). Further, this can also arise as the tagging period during JF might not always coincide with the period of lowest tropical cold point tropopause temperatures. Interannual variations, e.g. through the QBO, can induce anomalies on top of the mean annual

5    cycle (cf. e.g. Seidel et al., 2001; Kim and Son, 2012, for analyses of the mean annual cycle of tropical cold point tropopause temperatures). Consequently, the lowest temperatures and thus the strongest freeze-drying, might occur outside the tagging period.

Figures 7a and 7b show time series of water vapour mixing ratios for the individual source regions and for total water vapour

10   at 400 K in the NH extratropics ($50°N$–$70°N$) and the associated relative contributions to the water vapour and mass budget. For both, the absolute and relative contributions of water vapour of the specific source regions a small increase over time can be





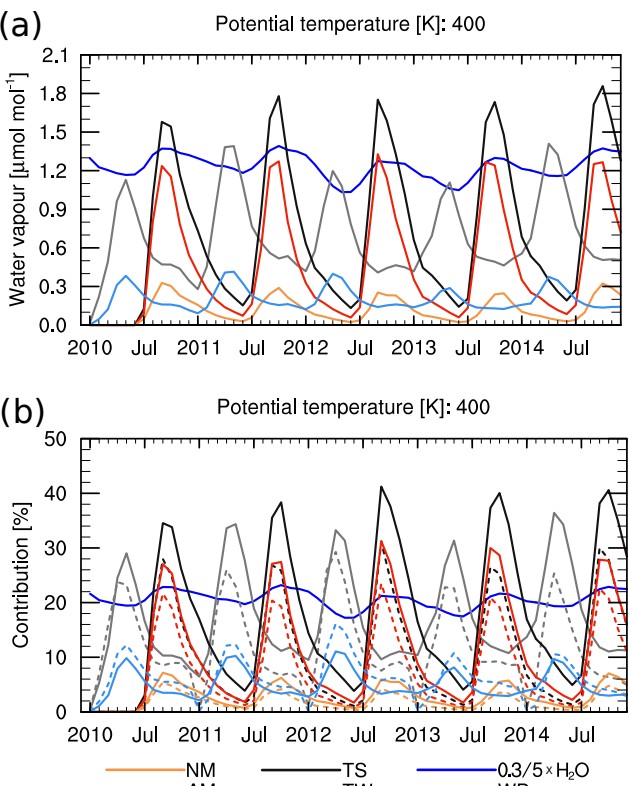

**Figure 7. (a)**: Time series of the $H_2O$ mean mixing ratios from the different source regions and water vapour (all in $\mu\text{mol mol}^{-1}$, total $H_2O$ scaled by $\times 0.3$). The time series are calculated over the region $50°$–$70°$N at 400K for the source regions TS (black), TW (grey), WP (light blue), AM (red) and NM (orange). Total water is shown in blue. **(b)**: Time series of the contribution of the mass (dashed) and $H_2O$ (solid) tracer for the different regions averaged over $50°$–$70°$N at 400 K. (Total $H_2O$ scaled by $\times 5$, where $1\%$ equals $1\mu\text{mol mol}^{-1}$). Colour coding as in **(a)**.

seen (especially for the TS and TW water vapour tracers) because the water vapour tracers are not set to zero prior to the next year's pulse (cf. Sect. 2.2). The average maximum absolute water vapour contributions from the AM and the TW tracer ($\sim$1.2–1.3 $\mu\text{mol mol}^{-1}$) to the NH extratropics at 400 K are similar and are roughly 3–4 times higher than the peak contributions from the WP water vapour tracer ($\sim$0.37 $\mu\text{mol mol}^{-1}$) and NM water vapour tracer ($\sim$0.29 $\mu\text{mol mol}^{-1}$), respectively. The absolute contributions of the TW water vapour tracer to the NH extratropics do not fall as low as the contributions from the TS tracer. The opposite is true for the TS and the TW tracer in the SH extratropics (not shown). This is probably related to the different strength of downward movement (experienced by the tracer) in accordance with the seasonal cycle of the BDC and the release period of the tracers (cf. also Fig. 2 of Ploeger and Birner, 2016). In the NH extratropics, apart from the air masses from the WP tracer, all tracers show a stronger water vapour than mass contribution indicating that relatively young air masses tend to moisten the NH extratropics. As this is already the case during the first year of the simulation, we can infer, that this is not due to not reinitialising water vapour. The average peak relative contribution to the water vapour budget in the NH extratropics



**Table 2.** Summary of average peak mass and water vapour contribution (in %) at 400 K (50°N–70°N) and 450 K (10°S–10°N).

| Source region | 400 K | | 450 K | |
|---|---|---|---|---|
| | mass | water vapour | mass | water vapour |
| AM | 22 | 29 | 12 | 14 |
| NM | 4.4 | 6.4 | 5.2 | 6.7 |
| TS | 28 | 39 | 36 | 51 |
| TW | 25 | 33 | 58 | 63 |
| WP | 12 | 9.8 | 30 | 24 |

from the AM tracer is roughly 29% (approximately twice as high as the contribution to the deep tropics, 10°S–10°S, at 450 K). In contrast, the WP and NM water vapour tracers show clearly lower relative contributions of ∼9.8% and ∼6.4%.

The complete set of average maximum contributions (in %) of the various regional tracers to the mass and water vapour
budget in the tropical pipe and the extratropics is summarised in Table 2.

### 3.4 Contribution to the tropical and extratropical tape recorder signal

The relative contribution of the AM and TS water vapour tracers to the water vapour budget in the inner tropics is shown in Fig. 8 as contours along with the water vapour anomalies (i.e. the tropical tape recorder), which are colour-coded. The contour lines of the contribution of the AM and the TS tracer nicely align with the positive water vapour anomalies of the tape recorder
signal. In contrast to the TS tracer, for the AM tracer the vertical position of the maximum contribution is not co-located with the maximum positive water vapour anomalies but somewhat higher (approximately in the height range 400–440 K). This is in agreement with the more poleward position of the AM source region in comparison to the TS source region and with the upward and equatorward movement of the AM water vapour (and air masses) as shown in Fig. 4. The longer transport pathway from the Asian monsoon region to the deep tropics leads to no obvious time lag between the moist phase of the tape recorder
signal and the maximum relative contributions of the AM water tracer. Further, the contributions of both water vapour tracers decay with height and time in agreement with the decay of the total $H_2O$ tape recorder signal, which indicates that these air masses are diluted gradually. As in the time series plot (cf. Fig. 6), the TS water vapour tracer clearly shows a larger relative contribution to water vapour in the tropical stratosphere than the AM tracer (e.g. averaged maximum contributions around 35% at 500 K compared to roughly 9%).

Figure 9 shows the horizontal evolution of the tape recorder signal at 400 K as water vapour anomalies with respect to the temporal and zonal mean. The relative contributions of the AM and the TS water vapour tracers are overlaid as black contours



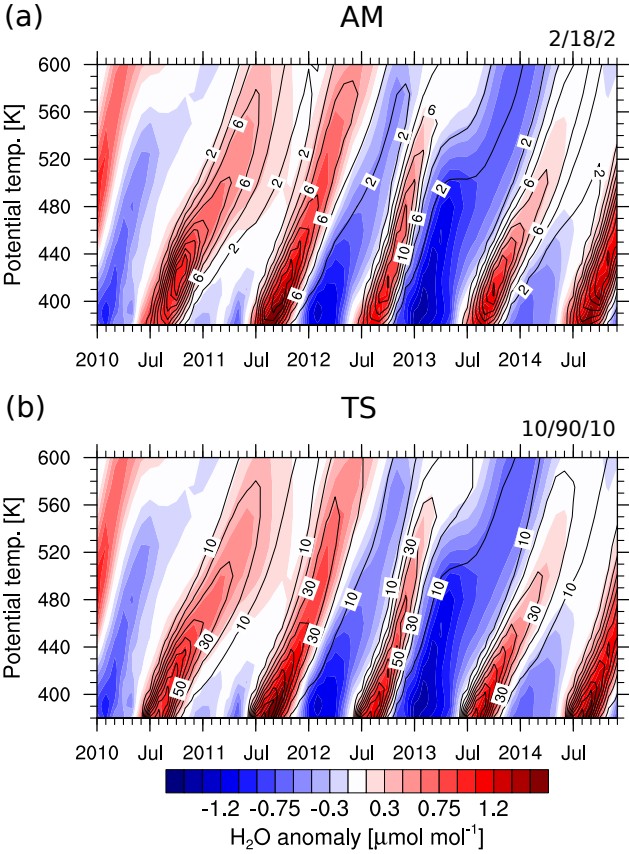

**Figure 8. (a)**: Vertical water vapour tape recorder (colour coded) as temporal anomalies for $10°$N–$10°$S (in $\mu$mol mol$^{-1}$) and corresponding relative contribution (in %) of the AM tracer to water vapour (black contours). Numbers in the top right corner indicate the contour line spacing as min/max/delta. **(b)**: As in **(a)** but for TS tracer.

(Figs. 9a and 9b, respectively). For the AM water vapour tracer the noticeable relative contributions to water vapour at 400 K are mostly located in the NH in agreement with the main transport pathway of monsoon air masses to the NH extratropics at these levels (cf. Fig. 4). At roughly $60°$ N the maximum relative contributions are typically $\sim$26–30% around September to October. The corresponding maximum relative contributions for the TS tracer at 400 K at $60°$ N are in the range of $\sim$33–40%

5     and thus only moderately higher than for the AM tracer. In agreement with Fig. 5, the TS water vapour tracer shows high contributions to water vapour also in the SH (average peak contribution of $\sim$30% at 400 K and $60°$ S). Remarkably, the highest relative contributions of TS water vapour are typically located somewhat south of the Equator. In particular during 2011–2013 the maximum is located roughly at $10°$ S.




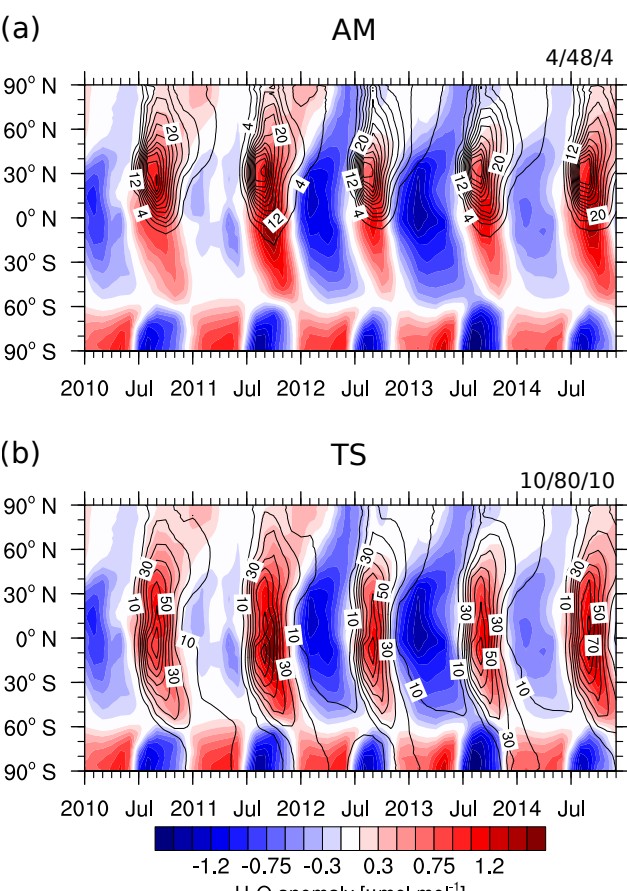

**Figure 9. (a)**: Horizontal water vapour tape recorder (colour coded) as temporal anomalies from the zonal means (in $\mu$mol mol$^{-1}$) and corresponding relative contribution (in %) of the Asian monsoon (AM) tracer to water vapour at 400 K (black contours). Numbers in the top right corner indicate the contour line spacing as min/max/delta. **(b)**: As in **(a)** but for TS tracer.





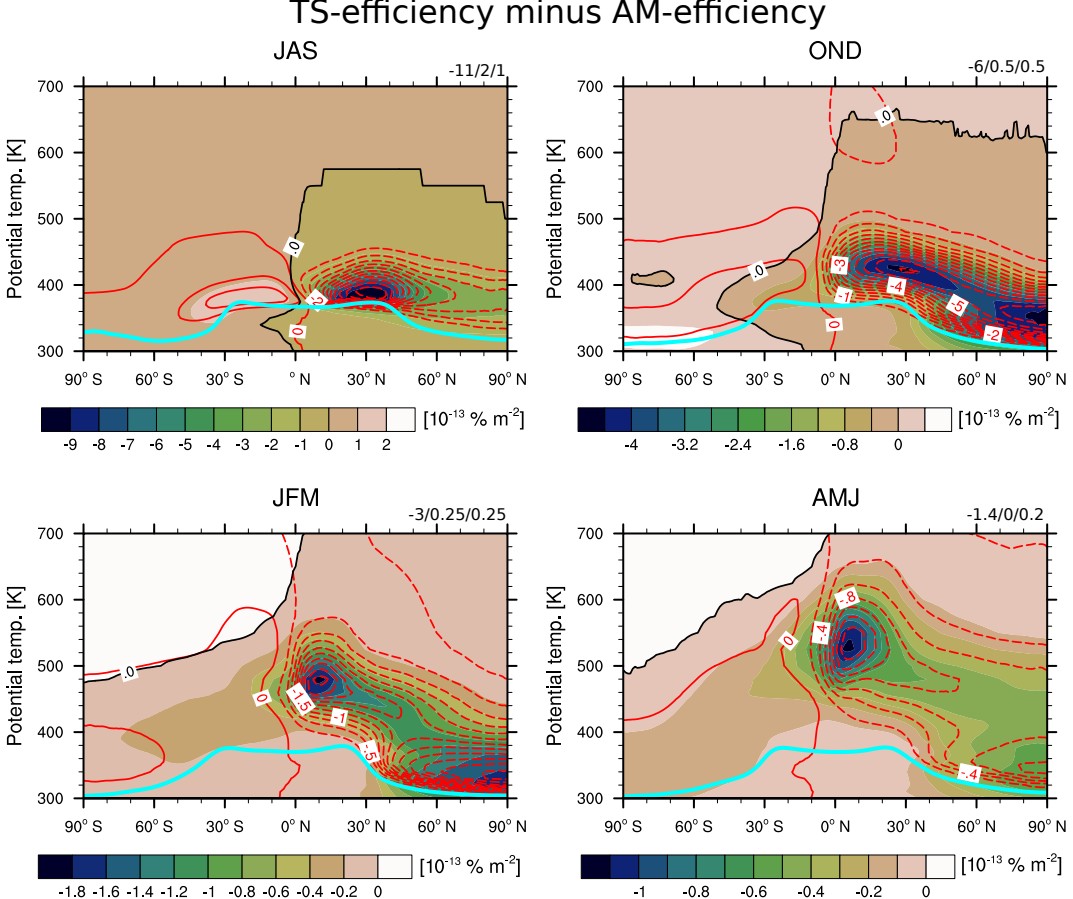

**Figure 10.** As in Fig. 5 but the difference of TS minus AM tracer contributions normalised by the respective source region area to yield the difference in efficiency (in $10^{-13}\,\%\,\mathrm{m}^{-2}$). Water vapour results are contoured in red (negative contours are dashed), whereas the mass tracer differences are colour-coded. The blue lines represent the mean WMO-tropopause based on ERA-Interim data. Note that the colourbar changes throughout the panels, the black line shows the zero contour of the mass tracer to guide the eye.

## 4 Discussion

The presented results may suggest that compared to the tropics during NH summer (TS tracer), the AM region plays a rather limited role for the tropical (10°S–10°N) stratospheric air mass and water vapour budget (mean peak contributions of 36% and 51% vs. 12% and 14% at 450 K, respectively; cf. also Table 2). Nevertheless, it first has to be mentioned, that the idealised mass tracers were initialized with unity in the respective source regions. However, if the transport of other trace gases which show high values within the monsoon anticyclone, such as CO (e.g. Park et al., 2007, their Fig. 5a), would be considered, this might considerably increase the importance of the AM region (cf. Randel et al., 2010, their Fig. 2). Secondly, we emphasise that the AM region is considerably smaller (cf. Table 1) than the tropical region and hence a smaller impact of the former on





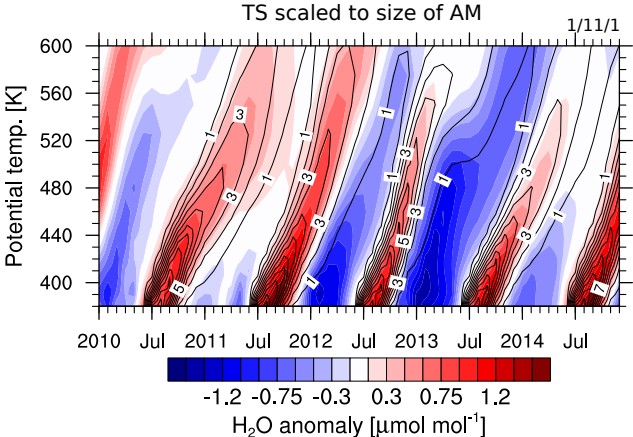

**Figure 11.** As in Fig. 8 but for the TS tracer rescaled by the ratio of the area AM divided by area TS.

the water vapour and mass tracer contribution to the tropical stratosphere can be expected. To make a comparison regarding the transport efficiency from the tropics during NH summer and the Asian monsoon region, we will normalize the contributions of the respective tracers by the size of the corresponding source region. In a similar way, the relative efficiency of aerosol transport from the Asian monsoon anticyclone to the stratosphere was compared to the transport through the tropics in Yu et al. (2017).

Figure 10 shows the difference of the efficiency, i.e. normalised by the size of the respective source region, of the TS minus AM tracers over the course of a year for water vapour (red contours) and mass contributions (colour-coded). Clearly, the AM tracers show higher efficiencies with respect to both water vapour and mass transport as the differences are mostly negative. Further, the patterns of the difference align closely with the original transport patterns for the AM mass and water vapour tracer as depicted in Fig. 4.

The contribution from the tropical water vapour tracer to the tropical tape recorder rescaled by the size of the AM region divided by the size of the TS region is shown in Fig. 11 (indicating the contribution of the TS tracer if the initialisation region was as large as the AM region). Consequently, Figs. 10 and the comparison of Figs. 8a and 11 shows that the AM tracer is more efficient with respect to mass and water vapour transport than the TS tracer.

Further, Fig. 12 shows the previously presented contributions of the different source regions to total water vapour and air mass
in the NH extratropics and the tropical stratosphere rescaled with the respective source region area, so the contribution/transport efficiency can be determined. With respect to the air mass tracers, the highest efficiency is found for the WP tracer for transport to the tropical pipe, followed by the AM tracer and the TW tracer. For transport to the NH extratropics the AM air mass tracer clearly shows the highest efficiency (cf. Fig. 12d).

With respect to water vapour the AM region is clearly the most efficient source region. Only the WP water vapour tracer
shows a similar efficiency for water vapour transport to the tropical stratosphere (cf. Fig.12a). Interestingly, the NM and the TS tracer show a comparable efficiency with respect to transport to the tropical stratosphere. The tracer released during NH winter (TW and WP, grey and light blue lines) show slower decreases in the contribution most likely related to the weaker upwelling



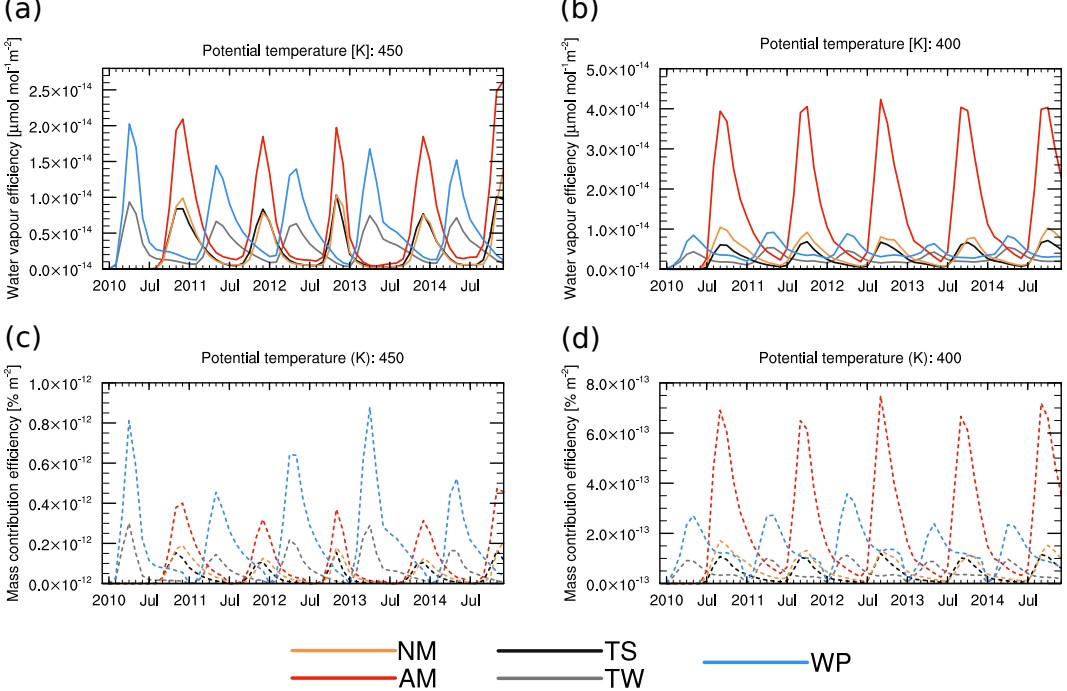

**Figure 12.** Time series of the $H_2O$ mean mixing ratios efficiency from the different source regions (all in $\mu mol\,mol^{-1}\,m^{-2}$) **(a)** averaged over the region $10°S$–$10°N$ at $450\,K$ and **(b)** averaged over the region $50°$–$70°N$ at $400\,K$. The colour coding of the source region is as follows: TS (black), TW (grey), WP (light blue), AM (red) and NM (orange). **(c)** and **(d)** as in **(a)** and **(b)** but for the mass contribution efficiency (in $\%\,m^{-2}$).

approximately half a year after the initialisation (cf. e.g. Abalos et al., 2012, their Fig. 3). Further, we analysed an additional passive water vapour tracer, which did not undergo freeze-drying after the initialisation. The corresponding results indicate, that the high efficiency of transport of water vapour to the stratosphere from the Asian monsoon region is mainly caused by the efficiency of the mass transport. Remarkably, freeze-drying of Asian monsoon air masses on their way to the stratosphere

5  strongly reduces the efficiency of the Asian monsoon in moistening the stratosphere compared to the tropics, i.e. air masses in the stratosphere from the Asian monsoon region compared to the tropics have experienced stronger dehydration. The high transport efficiency in the AM region has also been noted by Tissier and Legras (2016), who found that during NH summer convectively influenced air parcels from the AM region and in particular from the Tibetan Plateau are more likely to reach the 380 K level (approximately the tropopause height) than from other regions and periods (cf. their Fig. 2c).

10     To set our work into context, we refer to two previous studies that were targeted at investigating water vapour transport from the Asian monsoon region to the stratosphere. As noted in the introduction (Sect. 1) the studies from Bannister et al. (2004) and Wright et al. (2011), which assessed the contribution of the Asian monsoon (anticyclone) on the tropical stratospheric water vapour seemingly reached differing conclusions. Using a water vapour tagging approach Bannister et al. (2004) found that the ASM considerably contributes to the moist phase of the tape recorder. In detail, they calculated the drying or moistening effect





of water vapour from specific source regions by initialising water vapour in this region with previously simulated water vapour values while the rest of the tropics was imprinted with an annual mean stratospheric $H_2O$ value. Bannister et al. (2004) came to the conclusion roughly a quarter or more of the moist anomaly of the tape recorder is due to water vapour that is related to the Asian monsoon. Wright et al. (2011), on the other hand, used backtrajectories and determined the absolute moistening

effect of the monsoon region as the difference between the mean water vapour of all trajectories minus the mean water vapour of trajectories, that did not encounter convection in the Asian monsoon region. Based on this analysis, they argued that the Asian monsoon region only has a limited moistening effect. As the two scientific questions and the corresponding attribution methods differ – Bannister et al. (2004) assess the contribution to the wet anomaly, whereas Wright et al. (2011) assess the moistening effect of air from the Asian monsoon compared to the remaining air masses that entered the stratosphere (mostly)

during NH summer – discrepancies in the assessed attribution are explicable, as will be shown in the following.

Although the attribution method by Bannister et al. (2004) is similar to the tagging method used in this study, it differs due to their aim to assess the contribution to the wet phase of the tape recorder. In contrast, our tagging method allows to show the full contribution (absolute and relative) of water vapour from different source regions including the Asian monsoon. Further, compared to Bannister et al. (2004) our results are based on meteorological data from the ERA-Interim reanalysis, which is

observationally constrained and our model results do not show a shifted Asian monsoon water vapour signal at tropopause levels compared to satellite observations, which is present in Bannister et al. (2004, cf. their Fig. 3c). The use of reanalysis data also removes another bias, which seems to be present in the model results presented in Bannister et al. (2004): For their attribution calculation it seems that inert transport of water vapour is considered from the 100 hPa level further up into the stratosphere, in agreement with their reference model data, which shows a cold point tropopause below that pressure level

during NH summer in the monsoon region (cf. their Fig. 10b). This contradicts findings, e.g. by Pan et al. (2016; cf. their Fig. 1c), which show that the cold point tropopause in the AM region is located at lower pressures (higher altitudes).

Here, we also investigate the contribution of the Asian monsoon to stratospheric water vapour as investigated by Bannister et al. (2004) and Wright et al. (2011). In detail, analogous to the way described in the Appendix of Bannister et al. (2004) we calculate the contribution of the Asian monsoon to the wet phase of the tape recorder ($C_{B04}$) as:

$$C_{B04} = \frac{H_2O^{AM} + (1 - \chi^{AM}) \times \overline{H_2O} - \overline{H_2O}}{H_2O - \overline{H_2O}} = \frac{H_2O^{AM} - \chi^{AM} \times \overline{H_2O}}{H_2O - \overline{H_2O}}, \qquad (3)$$

where all quantities denote the spatial (10°S-10°N) and temporal mean over the respective period, here November–December (ND), at 450 K whereas the overbar denotes the 2010–2014 mean over the region at 450 K. The mass contribution of the Asian monsoon region is given as $\chi^{AM}$. For the analysis of the Asian monsoon contribution, $C_{B04}$ is calculated for each individual

year 2010–2014 over the period ND and then averaged. In our simulation this estimate of the Asian monsoon contribution yields roughly 26%, which is almost exactly the lower limit estimate given in Bannister et al. (2004), even though, the compared regions and the height level (somewhat lower in this study) differ from the setup in Bannister et al. (2004).





Following the description in Wright et al. (2011), we also calculate the moistening effect of the Asian monsoon ($C_{W11}$), according to the following formula:

$$C_{W11} = \frac{H_2O - \frac{H_2O - H_2O^{AM}}{1 - \chi^{AM}}}{H_2O},$$ (4)

where the notation is as above, i.e. means over 10°S-10°N at 450 K during ND are considered. Here, the denominator of
$1 - \chi^{AM}$ rescales the water vapour that was not influenced by the Asian monsoon ($H_2O - H_2O^{AM}$) to 100% air mass. Hence this fraction is the mean water vapour mixing ratio of air masses that were not affected by the Asian monsoon. As for the $C_{B04}$ contribution, here the $C_{W11}$ contribution is calculated for each individual year 2010–2014 over the period ND and then averaged. For our results, we find a moistening contribution of 2.6%, which is close to the ∼3% stated in Wright et al. (2011). Again, we note, that there are still differences between our setup and the one used in Wright et al. (2011), e.g. as Wright et al.
(2011) can only account for water vapour that has entered the stratosphere within the last three quarters of a year.

Although, we note that there are still differences in the details, e.g. the analysis period/height, considering mostly freshly entered air masses, etc. our analysis shows 1) that the results obtained here fit with previous model results and 2) that the initial scientific question and the corresponding attribution technique heavily influences the perceived importance of the Asian monsoon in determining stratospheric water vapour.

In Sect. 3.1 it was pointed out that our CLaMS-based results might be overestimating the influence of the Asian monsoon region on stratospheric water vapour as water vapour in the UTLS in the Asian monsoon region shows higher values than in satellite-based observations especially when compared with the results from the North American monsoon region (cf. Sect. 3.1). Hence, the results presented here might serve as an upper limit that could be realistic if our initialisation at 370–380 K plays no major role as freeze-drying will remove the memory of the initialisation. Further, the mass tracer results during summer could
be viewed as sensitivity, assuming an unweighted initialisation of water vapour (and no further freeze-drying). Also, there is mostly at least a factor two regarding the contribution of the AM versus the NM water vapour tracers, which is clearly higher than what could be expected from the difference at the top of the initialisation regions (mean water vapour values at 380 K of 6.2 $\mu$mol mol$^{-1}$ vs. 5.7 $\mu$mol mol$^{-1}$). However, we acknowledge, that the quantitative results presented here still depend on the model and the employed reanalysis data. Further, reanalysis data, including temperature data and the diabatic heating rates,
are another potential source of uncertainty in our calculations (Wright and Fueglistaler, 2013).

## 5   Conclusions

We now come back to address the main research goals of this study regarding water vapour transport from the Asian monsoon region to the stratosphere. These research goals have been stated in the introduction (Sect. 1) and are repeated here again. Our main tasks were:
1) To highlight the transport pathways of water vapour from the UT in the Asian monsoon to the stratosphere and to contrast air mass and water vapour transport from the Asian monsoon region to the stratosphere.

2) To quantify the impact of the Asian monsoon on the stratospheric water vapour budget.




3) To compare the water vapour and mass transport (efficiency) from the Asian monsoon to the transport (efficiency) from additional source regions, such as the North American monsoon and the entire tropics.

Based on our analysis of a multiannual CLaMS simulation with tagged mass and water vapour tracers we come to the following conclusions with respect to these research goals:

1) The bulk of water vapour from the UT in the Asian monsoon region is transported vertically into the lower stratosphere above the Asian monsoon region. Thereafter, Asian monsoon water vapour is either transported to the tropics, where it experiences further uplift or it is transported poleward and downward. Hence, water vapour from the UT in the Asian monsoon region is mostly determined by the transport pathways of air masses from the UT in the Asian monsoon in agreement with the BDC as previously described in Ploeger et al. (2017; cf. also Fig. 4 of this study).

2) Water vapour from the AM region contributes on average at most $0.65\,\mu\mathrm{mol\,mol^{-1}}$ (14%) to the water vapour in the tropical stratosphere at 450 K during the moist phase of the tape recorder. The average peak contribution to the NH extratropics at 400 K is considerably higher ($\sim 1.3\,\mu\mathrm{mol\,mol^{-1}}$ corresponding to 29%).

3) Compared to the NM region, the AM region shows higher peak mass and water vapour contributions both in the tropical stratosphere and extratropical lower stratosphere. The average maximum water vapour contribution to the deep tropics at 450 K of the AM ($0.65\,\mu\mathrm{mol\,mol^{-1}}$) is roughly twice as high as for the NM ($0.31\,\mu\mathrm{mol\,mol^{-1}}$) and almost comparable to the water vapour contribution of the TW tracer ($0.73\,\mu\mathrm{mol\,mol^{-1}}$). In the NH extratropics at 400 K the air masses from the Asian monsoon region show a high water vapour contribution of $1.3\,\mu\mathrm{mol\,mol^{-1}}$, which is comparable to the water vapour contribution from the NH winter tropics tracer ($1.2\,\mu\mathrm{mol\,mol^{-1}}$) and is only excelled by the NH summer tropics tracer ($1.7\,\mu\mathrm{mol\,mol^{-1}}$) in our study. With respect to mass and water vapour to the NH extratropics at 400 K, the AM tracers show the highest transport efficiency of all our tracers. Regarding the deep tropics at 450 K only the warm pool (WP) region shows a comparable transport efficiency with respect to water vapour and a higher efficiency with respect to air mass transport than the Asian monsoon region.

These results aim to better quantify the impact of the Asian monsoon (anticyclone) on the stratospheric water vapour budget. Further the results emphasise the efficiency of the monsoon region for transporting air masses and water vapour to the stratosphere.

*Data availability.* CLaMS data are available upon request from FP. MLS H$_2$O version 4.2 data were downloaded via the MLS website. These data are available from GES DISC (Lambert et al., 2015).

*Author contributions.* MN had the original idea (based on mass transport investigations from Garny and Randel, 2016, and Ploeger et al., 2017) and initiated the study. AP implemented water vapour tagging in CLaMS. AP and FP set up and performed the model experiments.



MN performed the data analysis and wrote main parts of the paper. All authors contributed to the study design, discussion of results and the writing of the paper.

*Competing interests.* The authors declare that no competing interests are present.

*Disclaimer.*

*Acknowledgements.* We thank Martin Dameris, Sabine Brinkop (both DLR) and Mengchu Tao (FZJ) for helpful discussions and support. Further, we thank Heidi Huntrieser (DLR) for thoughtful comments and remarks, which helped to improve the manuscript. We thank the ECMWF for providing ERA-Interim reanalysis data. We thank the MLS science team for the production of AURA-MLS data and its documentation. CDO (Climate Data Operators) have been used for data processing (available at http://www.mpimet.mpg.de/cdo). We used the NCAR Command Language (NCL; cf. also the list of references) for data analysis and graphics. NCL is developed by UCAR/NCAR/CISL/TDD. The

research leading to these results has received funding from the European Community's Seventh Framework Programme (FP7/2007 - 2013) under grant agreement n° 603557. AP and FP were funded by the Helmhotz Association under grant VH-NG-1128 (Helmholtz Young Investigators Group A-SPECi). This study has received funding by the Helmholtz Association under grant VH-NG-1014 (Helmholtz-Hochschul-Nachwuchsforschergruppe MACCClim). The work described in this paper has received funding from the Initiative and Networking Fund of the Helmholtz Association through the project "Advanced Earth System Modelling Capacity (ESM)". Additional funding came from the

DLR internal project KliSAW (Klimarelevanz von atmosphärischen Spurengasen, Aerosolen und Wolken).



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
