# Peer review of "Quantification of water vapour transport from the Asian monsoon to the stratosphere"

_Atmospheric Chemistry and Physics, 2019_

## Referee Comment (RC1) · Anonymous Referee #1 · 22 Mar 2019

General comments

This well-written paper describes the contribution of the Asian Monson to the water transport into the stratosphere. The paper is suitable for acceptance in ACP, once the authors address one general comment and several, minor, specific comments.

My general comments concerns the simulations on which the authors base their study are for the period Jan 2010 – Dec 2014. Is this a long enough period to provide reasonable, if not definitive, conclusions? I suggest the authors discuss this point at end of sect.4, where you describe weaknesses in the study, and in sect. 5.

Specific comments

[Figure]

P. 6

Table 1 caption: To help the reader, introduce what JA and JF mean.

L. 9: Perhaps provide more information on the differences between TWC(i) and TWC – from the text it seems to me that TWC represents the quantity for the whole tropics, and TWC(i) represents the quantity for the region under consideration.

P. 7

Fig.2: For this figure and similar figures, I suggest authors indicate what endpoints of colour scale show. E.g., red/blue high/low values of H2O.

P. 9

L. 13: I suggest the authors quantify this "excellent agreement".

P.13

L. 5: Could the authors speculate on the possible reasons of this result?

P. 14

Fig. 6: Remind the reader why you scale the total H2O by 0.3.

L. 1-2: I suggest probably -> likely, unless you can relate the phenomenon to a statistical distribution.

P. 15

Fig. 7: Remind the reader why you do the scaling.

P.16

Table 2: I suggest the authors highlight in bold the highest value(s).

L. 9: I suggest the authors avoid terms like "nicely". Maybe use "align well".

P. 17

L. 6: Why is this remarkable? Avoid subjective language.

P.19

L. 4-5: Perhaps the authors could provide more details of this explanation.

P. 20

L. 6: Do you need "clearly"? Omit needless words. Same elsewhere in the text.

L. 20: Why is this interesting? Avoid subjective language.

P. 21:

L. 4: Why is this remarkable? Avoid subjective language.

P. 24

L. 25: Perhaps mention the quality of the efficiency. Is it relatively high/low in itself? And compared to other studies?

P. 25

References: Check spelling of "O'Neill", including use of capitalization.

---

## Referee Comment (RC2) · Anonymous Referee #2 · 29 Mar 2019

This work aims to quantify the contribution of $H_2O$ transport from the Asian monsoon (AM) to the global stratosphere, based on calculations from CLaMS. Quantification of this influence is a long-standing question in the research community. The calculations are performed using tagged tracers from several different regions, including the monsoons, global tropics and the western Pacific. The model is compared to satellite observations and shown to simulate global stratospheric $H_2O$ in a realistic manner. The budget calculations are straightforward and the results seem reasonable, with the AM contributing $\sim$14% to the tropics during the summertime moist phase of the tape recorder, and $\sim$29% to NH summer high latitudes. The calculations also include estimates of the $H_2O$ transport 'efficiency', namely $H_2O$ scaled by mass transport, showing that the AM has relatively higher efficiency compared to other regions (because it is

moist to begin with, with strong transport pathways to the global stratosphere). The paper includes detailed comparisons to two previous publications (Bannister et al, 2004 and Wright et al, 2011) with apparently different conclusions, and explains the differing results as depending on the specific questions that are posed. Overall the calculations are clearly described, the paper is well written and the figures are clear and simple. The paper is appropriate for ACP, and this will be a well-referenced standard quantifying the monsoon contribution to global stratospheric H2O. This is an excellent paper – well done. I have only a few minor comments for the authors to consider:

1) P. 2, line 7: 'annual' seesaw instead of 'semiannual'

2) p. 13, line 30: omit 'supposed to be'

3) The units of efficiency are not very intuitive (10-13 or 10-14 % / m2 in Figs. 10 and 12). Could this be normalized to the area of the AM or global tropics (TS), to give a more physically meaningful value?

---

## Referee Comment (RC3) · Anonymous Referee #3 · 5 Apr 2019

This paper investigates the water vapor transport from the Asian monsoon region to the stratosphere using the ClaMS model with tagging method. The results are noteworthy, paper is well written, and figures are nicely generated. I recommend publication in ACP with minor revisions.

(1) In general, I was somewhat alarmed by the relatively dry NM at 83 hPa in the model compared to MLS (Fig. 2). The caveat is there so I am not suggesting any specific changes to the text, but the relative contributions of water vapor from AM and NM to the stratosphere, which are the key results of this study, have certain degree of uncertainty that is difficult to quantify.

(2) Comparison with Bannister et al. (2004) and Wright et al. (2011) studies is very nice. When reading the Introduction (specifically second paragraph of p. 3), I was also

reminded about the conclusion by Ueyama et al. (2018) that showed the importance of convection in explaining the water vapor maximum in AM at 100 hPa? Is this study not relevant for this paper because of its focus on the 100 hPa level? [Ueyama, R., Jensen, E. J., and Pfister, L. (2018). Convective influence on the humidity and clouds in the tropical tropopause layer during boreal summer. JGR, 123, 7576-7593.]

(3) I am slightly confused as to how the mixing scheme in CLaMS (i.e., merging of parcels and insertion of new parcels) affects the source identification through the tagging method. For example, if two parcels merge into one, does the merged parcel have two sources (if the two parcels have different sources)?

(4) Table 1: It may be helpful to spell out the acronyms (e.g., AM, NM) in the title.

(5) p. 9, line 13: I agree that the agreement is good, but the seasonal cycle amplitudes appear to be weaker in CLaMS compared to MLS. Any thoughts?

(6) p. 14, line 10: Why did you choose to show the time series at 400 K for the NH extratropics (Fig. 7) instead of at 450 K as for the tropics (Fig. 6)? It may be helpful to add "Tropics" and "NH extratropics" label in these two figures.

(7) p. 13, line 26: Should this be "14%" consistent with the Abstract (line 12) and Conclusions (line 11)?

(8) Table 2: The average peak mass and water vapor contributions from NM at 400 K and 450 K are very similar. I would have expected lower contributions at 450 K as for AM. Do you have an explanation?

(9) p. 24, line 8-9: This phrase, " Hence, water vapor from the UT in the Asian monsoon region is mostly determined by the transport pathways of air masses from the UT in the Asian monsoon", is unclear.

---

## Author Comment (AC1) · 28 May 2019

We thank the referees for the helpful comments on our manuscript. The supplement contains a detailed response to the points raised by the referees. At the end of the supplement the changes made between the discussion version and the revised version are highlighted.

Please also note the supplement to this comment:
https://www.atmos-chem-phys-discuss.net/acp-2019-169/acp-2019-169-AC1-supplement.pdf
* * *
[Figure]

2019.

**Supplement:**

**Author Comment on manuscript ACP-2019-169: "Quantification of water vapour transport from the Asian monsoon to the stratosphere"**

by M. Nützel et al.

May 28, 2019

We thank all referees for their helpful and encouraging comments on our manuscript. Below, we will address all of the points raised by the reviewers. The referees' comments are displayed in *black italics* and our reply is given in blue.

**Reply to comments from Referee #1 (ACPD, https://doi.org/10.5194/acp-2019-169-RC1, 2019)**

*Review of Nützel et al.*

*General comments*

*This well-written paper describes the contribution of the Asian Monson to the water transport into the stratosphere. The paper is suitable for acceptance in ACP, once the authors address one general comment and several, minor, specific comments.*

We thank referee#1 for taking the time to review our manuscript and are happy to receive positive feedback regarding our work. We will try to address all comments of the referee in a satisfactory way with our reply and corresponding changes in the manuscript.

*My general comments concerns the simulations on which the authors base their study are for the period Jan 2010 – Dec 2014. Is this a long enough period to provide reasonable, if not definitive, conclusions? I suggest the authors*

*discuss this point at end of sect.4, where you describe weaknesses in the study, and in sect. 5.*

We thank the referee for this helpful suggestion. Hence, we added a discussion of this point at the end of Sect. 4. The corresponding paragraph reads:
"The results presented here cover the summer monsoon periods from 2010 to 2014 and can be seen as an approximation to the climatological impact of the AM region on the stratospheric mass and water vapour budget. We emphasise that there is considerable interannual variability in the time series of the contributions from the different source regions (cf. Figs. 6 and 7). As an example, the annual peak mass contribution from the TS and AM region vary approximately between 27%–43% and 10–15%, respectively. Based on the water vapour anomalies displayed in Fig. 3 one could argue, that a large fraction of the interannual variability in stratospheric water vapour, e.g. caused by different QBO phases (cf. Sect. 3.1), is already covered by our simulation period. Still, it is possible that our results might not completely reflect the behaviour during specific years or time periods. As an example, Brinkop et al. (2016) argue that special constellations of the El Niño–Southern Oscillation (cf. e.g. Trenberth, 1997, and references therein) and QBO are responsible for the decrease of lower stratospheric water vapour in 2000. Such specific situations and associated possible changes in the dynamics might as well change the quantitative results of the attribution questions addressed here."
Further, in Sect. 5 we added that our results are based on simulations for the period 2010-2014, only. The correspdonding sentence before the list of our research results now reads:
"Based on our analysis of a multiannual CLaMS simulation covering 2010–2014 with tagged mass and water vapour tracers....".

*Specific comments*

*P. 6*
*Table 1 caption: To help the reader, introduce what JA and JF mean.*

JA and JF were previously introduced in the text, only. We now added a description below Table 1:
"The tagging periods are denoted with JF for January–February and JA for July–August."

64

*L. 9: Perhaps provide more information on the differences between TWC(i)*
*and TWC – from the text it seems to me that TWC represents the quantity*
*for the whole tropics, and TWC(i) represents the quantity for the region under*
*consideration.*

As noted by Referee#1, $\text{TWC}^i$ denotes the total water content from source
region $i$ (with $i$ being TS, TW, AM, NM or WP). TWC in turn represents total
water regardless of the source/tagging region. To be more precise regarding the
individual quantities and on how the tagging works, the technical description of
the tagging method (p.6 l.4-18 in the discussion manuscript) was restructured.
We hope that the method is easier to follow now. In particular the paragraph
introducing TWC and $\text{TWC}^i$ was rearranged and revised. It now reads:
"For technical reasons instead of water vapour, total water content (TWC), i.e.
water vapour ($H_2O$) plus ice water content (IWC), is being tagged and followed
along the parcels pathway. So, on each trajectory besides the information on the
"common" (i.e. not distinguishing between individual source regions) $H_2O$, IWC
and TWC tracers, the values of tagged total water from each of the individual
source regions, denoted by $\text{TWC}^i$ (with $i$ being one of the source regions, i.e.
$i \in \{\text{AM}, \text{NM}, \text{TS}, \text{TW}, \text{WP}\}$; cf. description of Table **??**), is available. The
mixing of the tagged total water tracers from the different source regions is
performed in analogy to the mixing of normal compounds, e.g. water vapour.
Hence, if one parcel is tagged in the AM and a second in the NM region (with all
other regional tracers being zero for each of the parcels) and these two parcels are
mixed, the resulting parcel contains non-zero values for $\text{TWC}^{\text{AM}}$ and $\text{TWC}^{\text{NM}}$.
Further, the evolution of tagged total water of a source region $i$ due to the
fallout of ice from time step $(t)$ to $(t + 1)$ on a certain trajectory is represented
as follows:

$$\text{TWC}^i(t+1) = \text{TWC}^i(t) \times \frac{\text{TWC}(t+1)}{\text{TWC}(t)}, \tag{1}$$

where $\text{TWC} = \text{IWC} + H_2O$ denotes the "common" total water content and the
superscript $i$ refers to the total water from source region $i$. Hence, $\text{TWC}^i$ is
assumed to change by the same percentage as TWC."

*P. 7*

*Fig.2: For this figure and similar figures, I suggest authors indicate what end-*
*points of colour scale show. E.g., red/blue high/low values of H2O.*

We added:

"$H_2O$ values above (below) 4.8 (3.4) $\mu$mol mol$^{-1}$ are shaded in dark red (blue)."
to the caption of Fig. 2. The captions of Figs. 3, 4, 8, 9 and 10 wers slightly
adjusted as well. Hopefully, the figures are now easier to read.

*P. 9*

*L. 13: I suggest the authors quantify this "excellent agreement".*

We thank the referee for this comment. We rephrased the part describing
the differences and agreement between MLS and CLaMS adding also informa-
tion on the correlation coefficients between CLaMS and MLS data. The relevant
part as in the revised version now reads:
"These figures display the anomalies with respect to the 2010–2014 mean water
vapour profile over the region 10°S–10°N. Apart from differences in the abso-
lute magnitude of the anomalies mostly around 100–70 hPa (higher anomalies in
MLS than in CLaMS with maximum anomalies of $\sim$1.6 $\mu$mol mol$^{-1}$ compared to
$\sim$1.4 $\mu$mol mol$^{-1}$, respectively) and slightly faster ascent in CLaMS than MLS,
the tape recorder signals from MLS and CLaMS data show excellent agreement
both with respect to the interannual variability and the strength of the $H_2O$
anomalies in the stratosphere (above $\sim$70 hPa). The correlation coefficients of
MLS and CLaMS monthly mean water vapour anomalies are above 0.8 for pres-
sure levels 100–56 hPa and 18–10 hPa and in the range of 0.64–0.75 for pressures
of 46–22 hPa. In particular, the occurrence of the driest phase ..."

*P.13*

*L. 5: Could the authors speculate on the possible reasons of this result?*

Following the referee's advice, we added the following sentence to better ex-
plain the reason of this result:
" Rather, the higher relative water vapour than mass contribution reflects the
fact that the tagged air masses (both TS and AM) are wetter than the comple-
mentary air masses encountered in the stratosphere."

*P. 14*

*Fig. 6: Remind the reader why you scale the total H2O by 0.3.*

We rephrased to:

"...total $H_2O$ scaled by $\times 0.3$ to fit the same scale as the regional tracers)"

*L. 1-2: I suggest probably -> likely, unless you can relate the phenomenon to a statistical distribution.*

Following the referee's advice to omit "probably", we changed from "probably" to "presumably" as "likely" is already used within this sentence.

*P. 15*

*Fig. 7: Remind the reader why you do the scaling.*

We rephrased to:

"...total $H_2O$ scaled by $\times 0.3$ to fit the same scale as the regional tracers)"

*P.16*

*Table 2: I suggest the authors highlight in bold the highest value(s).*

We thank the referee for this comment, however, we do not think that highlighting the highest values of each column would be helpful as then values from TS or TW only would be highlighted. From our point of view, this would be distracting. Maybe we did not understand the referee's intention...

*L. 9: I suggest the authors avoid terms like "nicely". Maybe use "align well".*

We have changed the wording according to the referee's advice.

*P. 17*

*L. 6: Why is this remarkable? Avoid subjective language.*

Done. "Remarkably" was deleted.

*P.19*

*L. 4-5: Perhaps the authors could provide more details of this explanation.*

We followed the referee's suggestion and rephrased to:

"However, the contribution might increase considerably when considering a

tracer with a non-homogenous source distribution. In this study the idealised mass tracer was initialised per definition with unity in the respective source regions, while realistic trace gases, such as CO exhibit a local maximum in the monsoon anticyclone (e.g. Park et al., 2007, their Fig. 5a), and thus the fraction of CO transported through the monsoon will be higher than estimated with the idealised tracer here. This might considerably increase the importance of the AM region (cf. Randel et al., 2010, their Fig. 2, which highlights transport through the Asian monsoon region). Further, we emphasise that the AM region is substantially smaller"

*P. 20*

*L. 6: Do you need "clearly"? Omit needless words. Same elsewhere in the text.*

"Clearly" was removed here as suggested by the referee.

*L. 20: Why is this interesting? Avoid subjective language.*

The wording was changed. The sentence now starts with: "We note, that...".

*P. 21:*

*L. 4: Why is this remarkable? Avoid subjective language.*

Following the referee's advice, "Remarkably" was omitted.

*P. 24*

*L. 25: Perhaps mention the quality of the efficiency. Is it relatively high/low in itself? And compared to other studies?*

We thank the referee for this comment and we would be happy to include a discussion if the referee has some references in mind. However, so far we do not know of previous studies that address water vapour or mass transport efficiency as investigated here for different source regions. As discussed e.g. in the introduction of the discussion paper, Yu et al. (2017) looked at the efficiency with respect to the contribution of aerosols affected by the monsoon region compared to the tropics. Furthermore, we do not keep track of how much air masses are actually tagged (instead our efficiency is given with respect to the area), so a

budget analysis as e.g. in Garny and Randel (2016) is not possible either. From our point of view, the best way to make use of the efficiencies is to compare them with the efficiency of a reference region. In our case this would be the tropics in summer, i.e. the results of the TS tracer.

*P. 25*

*References: Check spelling of "O'Neill", including use of capitalization.*

Done.

**Reply to comments from Referee #2 (ACPD, https://doi.org/10.5194/acp-2019-169-RC2, 2019)**

*This work aims to quantify the contribution of H2O transport from the Asian monsoon (AM) to the global stratosphere, based on calculations from CLaMS. Quantification of this influence is a long-standing question in the research community. The calculations are performed using tagged tracers from several different regions, including the monsoons, global tropics and the western Pacific. The model is compared to satellite observations and shown to simulate global stratospheric H2O in a realistic manner. The budget calculations are straightforward and the results seem reasonable, with the AM contributing ∼14% to the tropics during the summertime moist phase of the tape recorder, and ∼29% to NH summer high latitudes. The calculations also include estimates of the H2O transport "efficiency", namely H2O scaled by mass transport, showing that the AM has relatively higher efficiency compared to other regions (because it is moist to begin with, with strong transport pathways to the global stratosphere). The paper includes detailed comparisons to two previous publications (Bannister et al, 2004 and Wright et al, 2011) with apparently different conclusions, and explains the differing results as depending on the specific questions that are posed. Overall the calculations are clearly described, the paper is well written and the figures are clear and simple. The paper is appropriate for ACP, and this will be a well-referenced standard quantifying the monsoon contribution to global stratospheric H2O. This is an excellent paper – well done. I have only a few minor comments for the authors to consider:*

We thank the reviewer for taking time to serve as a referee for our manuscript. Further we are happy to receive such positive feedback with respect to our manuscript. The minor comments of referee#2 will be addressed here and in the revised version (when appropriate).

Regarding the transport "efficiency", we noticed, that there is a misunderstanding of what the efficiency is actually showing. The efficiency here is simply the mass contribution or the mixing ratio scaled by the area size of the respective source region. To avoid such a misunderstanding we have rephrased some key sentences so that the "efficiency" is described more clearly. Regarding the research tasks (Sect. 1 and Sect. 5) the last item was rephrased:

"To compare the water vapour and mass transport and the corresponding transport efficiencies from the Asian monsoon to the transport and transport efficiencies from additional source regions, such as the North American monsoon and the entire tropics."

And also in the discussion, where the efficiency is introduced we have rephrased to:

"We also compare the transport efficiency from the tropics during NH summer to the transport efficiency from the Asian monsoon region. For that purpose, we define the "efficiency" as the mixing ratio or mass contribution from a source region normalised by the corresponding source area. Yu et al. (2017) used a similar definition of transport efficiency (ibid. also the restricted lifetime of the anticyclone was taken into account) to assess the efficiency of aerosol transport from the Asian monsoon anticyclone to the stratosphere. Figure 10 shows the difference of the efficiency for water vapour (red contours) and mass contributions (colour-coded), i.e. the difference of the water vapour and mass contribution normalised by the size of the respective source region, of the TS minus AM tracers over the course of a year."

*1) P. 2, line 7: "annual" seesaw instead of "semiannual"*

Done.

*2) p. 13, line 30: omit "supposed to be"*

Done.

*3) The units of efficiency are not very intuitive (10-13 or 10-14 % / m2 in Figs. 10 and 12). Could this be normalized to the area of the AM or global tropics (TS), to give a more physically meaningful value?*

We agree with the referee's point of view. In fact, we have thought about scaling the results as suggest by the referee (so units would be %) when preparing these figures for the discussion paper. However, when rescaling the AM contribution to the size of the TS region the percentages could be above 100% (see e.g. Fig. 4 top right panel showing OND mass contributions of up to 16% which would correspond to roughly 133% when rescaled); meaning if the AM region was as large as the TS region the contribution would be above 100%, which would look even more odd. To avoid this, the tape recorder in Fig. 11

shows the TS contribution rescaled by the size of the AM region. Hence, we are in favour of sticking with these "unintuitive" units which however can be multiplied by the size of the AM and TS region to give the full contribution as given in the corresponding figures, which were presented in Sect. 3 of the discussion paper. In the caption of Fig. 12 we added the following sentence: "Multiplication of the efficiencies with the size of the respective source regions yields the contributions as shown in Figs. 6b and 7b."

**Reply to comments from Referee #3 (ACPD, https://doi.org/10.5194/acp-2019-169-RC3, 2019)**

*This paper investigates the water vapor transport from the Asian monsoon region to the stratosphere using the ClaMS model with tagging method. The results are noteworthy, paper is well written, and figures are nicely generated. I recommend publication in ACP with minor revisions.*

We thank referee#3 for the comments on our manuscript and for constructive feedback. The revisions asked for by referee#3 are addressed below and at appropriate places in the revised manuscript.

*(1) In general, I was somewhat alarmed by the relatively dry NM at 83 hPa in the model compared to MLS (Fig. 2). The caveat is there so I am not suggesting any specific changes to the text, but the relative contributions of water vapor from AM and NM to the stratosphere, which are the key results of this study, have certain degree of uncertainty that is difficult to quantify.*

We agree with the referee's classification of the NM monsoon being to dry in the model simulation compared to MLS and with the statement, that the impact of this bias is hard to assess. However, we note that we have tried to make such an estimation, e.g. by referring to the results of the mass transport calculations (see p. 23 l. 15-24 in the discussion paper) and we thoroughly address this caveat in the discussion and note it at several instances in the paper. Additionally, we changed the last paragraph of the Conclusion section, which now reads:

"These results aim to better quantify the impact of the Asian monsoon (anticyclone) on the stratospheric water vapour budget and although, the quantitative results are to some degree depending on the model and the tagging approach, we expect the qualitative results to be robust. Further the results emphasise the efficiency of the monsoon region for transporting air masses and water vapour to the stratosphere."

Further, we note that it seems to be difficult to get all aspects of the AM and NM water vapour signals "right" in model simulations (see p.9 l.1-9 in the discussion paper).

*(2) Comparison with Bannister et al. (2004) and Wright et al. (2011) stud-
ies is very nice. When reading the Introduction (specifically second paragraph of
p. 3), I was also reminded about the conclusion by Ueyama et al. (2018) that
showed the importance of convection in explaining the water vapor maximum in
AM at 100 hPa? Is this study not relevant for this paper because of its focus on
the 100 hPa level? [Ueyama, R., Jensen, E. J., and Pfister, L. (2018). Convec-
tive influence on the humidity and clouds in the tropical tropopause layer during
boreal summer. JGR, 123, 7576-7593.]*

We are pleased that the referee likes our comparison with the results from
Bannister et al. (2004) and Wright et al. (2011). We are glad to refer to the
suggested literature at appropriate places in the text, i.e. we have put a refer-
ence to the paper:
(1) the following sentence was added to the introduction (behind paragraph 2
on page 3 of the discussion paper):
"On the other hand, Ueyama et al. (2018) have employed a trajectory model
and observed convective cloud top data and demonstrated that convection is
a key element for producing high water vapour values in the Asian monsoon
anticyclone at $100\,\mathrm{hPa}$".
(2) in Sect. 3.1, where the differences between MLS and ClaMS with respect to
water vapour at $83\,\mathrm{hPa}$ are discussed, we rephrased to:
"Hence, modelling realistic water vapour distributions, in particular in the North
American and Asian monsoon region, is challenging (cf. e.g. Ueyama et al., 2018;
Wang et al., 2018, and references therein)."

*(3) I am slightly confused as to how the mixing scheme in CLaMS (i.e.,
merging of parcels and insertion of new parcels) affects the source identification
through the tagging method. For example, if two parcels merge into one, does
the merged parcel have two sources (if the two parcels have different sources)?*

In light of referee#1's question regarding TWC and $\mathrm{TWC}^i$ (the specific com-
ment regarding p.6 l.9), the technical description of the tagging method (p.6
l.4-18 in the discussion manuscript) was revised. We hope that this additional
information together with the information about the mixing also answers the
present question by referee#3. In short: each of the tagged total water tracers
($\mathrm{TWC}^i$) is regarded as a different tracer and mixed like a usual CLaMS tracer.
So after mixing (but also during initialisation, as the region TS comprises e.g.

parts of the AM region, cf. p.6 l.15-18 in the discussion manuscript) $TWC^i$ from two or more different source regions might be larger than zero indicating that (fractions of) the air parcel can be attributed to different source regions.

(4) Table 1: It may be helpful to spell out the acronyms (e.g., AM, NM) in the title.

We followed the referee's advice and in addition to the description in Fig. 1, the naming of the source regions was added to the description below Table 1 in the revised manuscript.

(5) p. 9, line 13: I agree that the agreement is good, but the seasonal cycle amplitudes appear to be weaker in CLaMS compared to MLS. Any thoughts?

We thank the reviewer for pointing out this difference in CLaMS and MLS data. According to referee#1's comment on the tape recorder signal we rephrased the part describing the MLS and CLaMS water vapour tape recorder signals in the revised version. Here, the statement that the tape recorder is weaker in CLaMS is included. As to the reasons for the weaker tape recorder signal, we can simply make some speculations about what might be the cause: apart from a too weak representation in the model (e.g. it could be that there is too much mixing in CLaMS in the tropics), this could also be partly caused from the fact that we do not apply apriori profiles to CLaMS data (cf. Sect. 3.1).

(6) p. 14, line 10: Why did you choose to show the time series at 400 K for the NH extratropics (Fig. 7) instead of at 450 K as for the tropics (Fig. 6)? It may be helpful to add "Tropics" and "NH extratropics" label in these two figures.

Figures 6, 7 and 12 have been changed according to the referee's advice. We chose 450 K for the tropics as this is clearly above the tropopause and at the lower edge of the tropical pipe and should reflect air masses that made it well into the stratosphere. For the extratropics 400 K is clearly within the stratosphere, further, the shallow branch of the BDC is expected to transport air masses from the tropics to the extratropics rather horizontally. Figures 4 and 5 show exactly this behaviour of the mass and water vapour tracers from the AM and TS region. In Ploeger et al. (2017) these two transport pathways are described to be "slower" and "fast", respectively. The contributions at 450 K

in the extratropics are rather limited as most horizontal exchange occurs below this level (see again Figs. 4 and 5).

*(7) p. 13, line 26: Should this be "14%" consistent with the Abstract (line 12) and Conclusions (line 11)?*

15% is correct here as this value refers to the mass contribution, whereas in the sentence thereafter the contributions to the water vapour budget are given. For the water vapour contribution 14% is stated as in the abstract. We adjusted the sentence and now refer to Table 2 also for the mass results:
"During the simulated years the mass contribution of the Asian monsoon tracer reaches at maximum 15% with an average maximum contribution of 12% (cf. Table 2)."

*(8) Table 2: The average peak mass and water vapor contributions from NM at 400 K and 450 K are very similar. I would have expected lower contributions at 450 K as for AM. Do you have an explanation?*

We thank the reviewer for noting this difference between the AM and NM region. When looking at the transport of air masses and water vapour from the NM region in analogy to Fig. 4, where the transport from the AM region is shown, we find the following: During JAS for the NM tracers the maximum is not located latitudinally within the centre of the initialisation region as for the AM tracers. We assume that this is related to the difference in the maximum altitude of the anticyclonic circulations in the AM and NM region (cf. e.g. Dunkerton, 1995; Gettelman et al., 2004). From our point of view, it seems reasonable that for the NM region horizontal transport occurs at lower levels than 400 K, whereas because the anticyclonic circulation extends higher in the AM region here also more horizontal transport at higher levels to the extratropics is facilitated, e.g. through eddy shedding (see e.g. Popovic and Plumb, 2001).

*(9) p. 24, line 8-9: This phrase, " Hence, water vapor from the UT in the Asian monsoon region is mostly determined by the transport pathways of air masses from the UT in the Asian monsoon", is unclear.*

We thank the referee for noting this unprecise statement. We rephrased the sentence and hope that it is easier to understand now:

445 "Hence, water vapour transport from the UT in the Asian monsoon region to
446 the stratosphere closely follows the pathways of mass transport (cf. Fig. 4). The
447 mass transport in turn, is in agreement with transport within the BDC as pre-
448 viously described in Ploeger et al. (2017; cf. also Fig. 4 of this study)."

[revised manuscript text omitted]